# Small Molecule Inhibitors for Hepatocellular Carcinoma: Advances and Challenges

**DOI:** 10.3390/molecules27175537

**Published:** 2022-08-28

**Authors:** Monica A. Kamal, Yasmine M. Mandour, Mostafa K. Abd El-Aziz, Ulrike Stein, Hend M. El Tayebi

**Affiliations:** 1The Molecular Pharmacology Research Group, Department of Pharmacology, Toxicology and Clinical Pharmacy, Faculty of Pharmacy and Biotechnology, German University in Cairo-GUC, Cairo 11835, Egypt; 2School of Life and Medical Sciences, University of Hertfordshire Hosted by Global Academic Foundation, New Administrative Capital, Cairo 11578, Egypt; 3Experimental and Clinical Research Center, Charité-Universitätsmedizin Berlin and Max-Delbrück-Center for Molecular Medicine, 13125 Berlin, Germany

**Keywords:** small molecule inhibitors, growth factor receptors inhibitors, tyrosine kinase inhibitors, hepatocellular carcinoma, molecular targets, small molecules as immunomodulators, HCC pathways inhibitors

## Abstract

**Highlights::**

Multi tyrosine kinase inhibitors licensed for HCC treatment.Multi kinase inhibitors not licensed for HCC treatment.Inhibitors of Growth Factor Receptors.Small molecules acting as immunomodulators.Small molecules inhibiting crucial HCC pathways.Small molecules targeting various molecular targets.

**Simple Summary:**

Liver cancer is one of the most common types of cancer globally. Its treatment options have been limited. Sorafenib was the most commonly used drug with patients that have advanced liver cancer. Recently, multiple new target proteins and pathways, which play a major role in the disease, have been discovered. Accordingly, researchers have designed and revealed new drugs. Some of them are FDA approved and others are under investigation in clinical trials. The aim of our systematic review is to provide an overview of these recently reported targets and compounds. This review will be useful in identifying unpopular or underrated targets, as well as designing new combination treatment strategies.

**Abstract:**

According to data provided by World Health Organization, hepatocellular carcinoma (HCC) is the sixth most common cause of deaths due to cancer worldwide. Tremendous progress has been achieved over the last 10 years developing novel agents for HCC treatment, including small-molecule kinase inhibitors. Several small molecule inhibitors currently form the core of HCC treatment due to their versatility since they would be more easily absorbed and have higher oral bioavailability, thus easier to formulate and administer to patients. In addition, they can be altered structurally to have greater volumes of distribution, allowing them to block extravascular molecular targets and to accumulate in a high concentration in the tumor microenvironment. Moreover, they can be designed to have shortened half-lives to control for immune-related adverse events. Most importantly, they would spare patients, healthcare institutions, and society as a whole from the burden of high drug costs. The present review provides an overview of the pharmaceutical compounds that are licensed for HCC treatment and other emerging compounds that are still investigated in preclinical and clinical trials. These molecules are targeting different molecular targets and pathways that are proven to be involved in the pathogenesis of the disease.

## 1. Introduction

Hepatocellular carcinoma (HCC) represents up to 90% of all liver malignancies [1]. It presents a tremendous global burden, being the sixth most common cancer and the fourth leading cause of all cancer deaths, accounting for 8.2% of cancer deaths, as reported by World Health Organization (WHO) in 2018. It has been reported that 80–90% of HCC cases arise in the context of chronic liver diseases (CLD). This can be attributed to a multitude of etiological risk factors and combination of more than one risk factor would have an incremental effect in HCC development [2,3,4]. These risk factors can be classified into seven major categories: hepatotropic viruses, such as HBV, HCV, and HDV; fatty liver disease, whether it is alcoholic or non-alcoholic; genetic and related factors; lifestyle factors (alcohol consumption, smoking, and dietary factors); environmental toxins (e.g., aflatoxins and contaminated water); metabolic diseases (diabetes and obesity); and miscellaneous factors (age, sex, epidermal growth factor polymorphism, gallstones, and cholecystectomy) [3].

In the last decade, several therapeutic modalities have proved to have a positive impact on HCC. These therapeutic modalities are divided into surgical therapies (ST) (resection, cryo-ablation, and liver transplantation) and non-surgical therapies (NST), which could be directed to liver (i.e., percutaneous ethanol injection, radiofrequency/microwave ablation, trans-arterial embolization, radiation therapy) or systemic (chemotherapy, molecularly targeted therapy and immunotherapy) [5]. The management of HCC involves a complex decision-making process, based on the tumor burden (total number of HCC lesions, diameter of the biggest target lesion, and total diameter of all the target lesions), degree of metastasis, hepatic decompensation asperity, comorbidities, and cancer related symptoms, as well as the patients’ desire to manage the condition [6]. In addition, the availability of treatment options is highly variable between medical centers in different countries with various levels of expertise and resources. Thus, HCC management requires a multi-disciplinary team approach to achieve the best outcome [7].

Surgical resection and transplantation are very effective in early stages of HCC, and the survival rate for 5 years is more than 70% [1]; however, advanced HCC patients represent more than 20% of the HCC patients in spite of surveillance programs [2]. Ablation or trans-hepatic arterial chemoembolization (Loco-regional therapies) are applicable to liver-limiting diseases. For patients diagnosed with extra-hepatic disease or non-responders to loco-regional therapy, systemic therapy (Molecularly Targeted therapy and Immunotherapy) is used [3]. In this review, we will be discussing the different small molecule inhibitors targeting various molecular targets that were confirmed to have major roles in the pathogenesis of the disease.

## 2. Methods

Research was performed at the States National Library of Medicine (PubMed). “Hepatocellular carcinoma”/“liver cancer”, “small molecule inhibitors”, and “treatment” were the descriptors used during the search process. The identified records from the search process included original research papers, book chapters, and reviews. These records were screened according to their relevance to the aim of the review and summarized. Research studies that discussed HCC therapy modalities other than small molecule inhibitors, such as monoclonal antibodies, miRNA-based therapies, or traditional approaches, were not included. Additionally, papers that focused on employing anti-viral medications to treat hepatitis, as well as papers that discussed drug delivery techniques, such as nanoparticles, were excluded. Finally, case reports unrelated to the aim of the review were removed during the screening process. Inclusion criteria were: complete English publications, chiefly discussing emerging small molecule inhibitors or possible combinational therapies to enhance the performance of previously known small molecules, which are available online, published between 2010 and 2021, with detailed information about contributors, methods, and analyzed results. Data collection was done during March/April 2021. Internal validity rather than the conclusion was the basis for bias elimination (Figure 1).

## 3. Results

It is worth mentioning that, over the last decade, there has been a significant number of publications reporting the identification of novel small molecule inhibitors in HCC, which indicates a growing interest in this therapeutic strategy among researchers, due to its many advantages over the conventional treatment modalities (Figure 2).

## 4. Tyrosine Kinases Inhibitors

### 4.1. Multi-Kinase Inhibitors Currently Licensed for HCC

#### 4.1.1. Sorafenib (BAY 43-9006)

Sorafenib is an oral bi-aryl urea drug, which is known for its ability to block several crucial oncogenic signaling pathways. It exerts its mechanism of action through inhibiting the fetal liver tyrosine kinases receptor 3 (FLT-3); vascular endothelial growth factor receptor (VEGFR)-1, -2, and -3; serine-threonine kinases c-Raf (Raf-1) and B-Raf; the cytokine receptor c-Kit; platelet-derived growth factor receptors (PDGFR) α and β; the MAPK/ERK kinase (MEK); and the rearranged during transfection (RET) receptor tyrosine kinase [8,9,10]. The mechanism of action is exerted by interaction with the ATP binding site of the targets [8]. Sorafenib was accepted by FDA for treating of advanced HCC patients with BCLC stage C and Child-Pugh A cirrhosis in 2007 (Figure 3) based on two major phase III clinical trials [11,12,13]. However, the clinical value of sorafenib is offset due to sorafenib resistance [14]. In this review, we report different strategies used to improve this resistance, including the combination of sorafenib with other small molecules in pre-clinical stages and recent clinical trials.

##### Interferon-Lambda 3

A combination of Interferon-lambda 3 (IFN-l3), along with sorafenib, has been reported to enhance the response to sorafenib. IFN-l3 is a type III interferon with anti-viral, anti-proliferative, and immunomodulatory functions [14]. A study demonstrated that the viability of liver cancer cell lines, HepG2 and SMMC7721, was suppressed by using a combination of IFN-l3 and sorafenib, in CCK-8 and colony formation assays, more than treatment with either alone. In addition, this combinational treatment promoted the loss of mitochondrial membrane potential and induced the production of ROS more than treatment with either alone, as shown by the flow cytometry. Furthermore, using a subcutaneous SMMC7721 tumor model, treatment with a combination of IFN-l3 and sorafenib significantly reduced the tumor growth/volume and induced apoptosis compared to treatment with sorafenib alone. Accordingly, IFN-l3 and sorafenib combination facilitates a synergistic effect on suppressing HCC cancer growth and promoting cell apoptosis in-vitro and in-vivo [14] (Figure 4).

##### Pregnenolone, Lomustine, Carisoprodol, Prestwick-1100, Chlorambucil, and Bretylium Tosilate

Another study employed Bioinformatic methods to discover new agents that could overcome Sorafenib resistance. To achieve this purpose, the gene expression profiles of cell lines with acquired sorafenib resistance, HCC-3sp and sorafenib sensitive HCC-3p, were obtained from an online database (Gene Expression Omnibus). In total, 541 differentially expressed genes (DEGs) were then identified and selected using a software (dChip) and their functions were analyzed by pathway enrichment and Gene Ontology (GO) analysis. These DEGs were associated with multiple pathways including cell adhesion and binding-related items. In addition, eight dysfunctional pathways that confer sorafenib resistance (adhesion and metabolism-related pathways) were enriched as indicated by KEGG pathway analysis. Then, the Connectivity Map was utilized to predict potential chemicals for reversing sorafenib resistance. Finally, several small molecules (pregnenolone, lomustine, carisoprodol, Prestwick-1100, chlorambucil, and bretylium tosilate) were screened out by using the CAMP tool. They had negative enrichment scores acting as potential therapeutic agents capable of overcoming sorafenib resistance [15] (Figure 4).

##### MLN4924

MLN4924 is a small molecule inhibitor of NEDD8-activating enzyme (NAE-1). NAE-1 plays a crucial role in protein neddylation, which is an additional dimension of sorafenib resistance and was reported to be abnormally activated in various types of cancers. Neddylation is the process by which the ubiquitin-like protein NEDD8 is conjugated to its target proteins. Proteins are targeted for degradation within the ubiquitin–proteasome system (UPS) through three steps involving 3 enzymes: (1) ubiquitin is activated into ubiquitin adenylate through ATP addition by ubiquitin-activating enzyme (E1); (2) ubiquitin adenylate is then transferred to the ubiquitin-transferring enzyme (E2) by forming a thioester bond; and (3) the third enzyme, ubiquitin ligase (E3), stimulates ubiquitin transfer from E2 to the Lysine residue of substrates [16]. The third enzyme in this cascade (Ubiquitin ligase) belongs to the large family of ligases that are called cullin-RING ligases (CRLs) [17]. To activate CRLs, these enzymes need covalent binding of NEDD8 to cullin proteins by (NAE) [18]. Accordingly, NAE inhibition by small molecules would inhibit CRL-mediated UPS. Protein neddylation components, NEDD8 and NAE1, are highly expressed in HCC and were associated with poor survival of patients. MLN4924 alone was reported to significantly inhibit the viability of HCC cell lines, increased apoptosis, and reduced migration capacity. However, in-vitro and in-vivo assays demonstrated that combining MLN4924 with sorafenib appears to have a synergistic effect on treatment of HCC. In-vitro assays showed that MLN4924 at a low concentration considerably increased the inhibition of cell proliferation and migration, as well as enhancing sorafenib-induced apoptosis. In-vivo experiments (xenograft mouse HCC models) confirmed that MLN4924 increased the anti-tumor efficacy of sorafenib via upregulation of cullin-RING E3 ubiquitin ligase along with its substrates: p21, p27, IκBɑ and Deptor. These results, taken together, propose that combination therapy of MLN4924 with sorafenib seems to have a synergistic effect on HCC treatment [19] (Figure 4).

##### Luteolin

Luteolin is a flavonoid found naturally in a range of vegetables. Luteolin was found to be involved with multiple pathways associated with cancer. It was reported that it induces apoptosis, anti-angiogenic effects, and cell cycle arrest [20,21,22]. In addition, luteolin is associated with protein kinases inhibition and redox regulation, reflecting its pro-apoptotic nature [23]. Additionally, luteolin sensitizes cancer cells to cytotoxicity induced by therapy through stimulating apoptosis pathways and inhibiting cell survival pathways [24,25,26]. A study investigated the effect of combination therapy of sorafenib along with luteolin. Results demonstrated that this combination synergistically induced cytotoxicity and apoptosis of HCC cell lines. Apoptosis potentiation was demonstrated by the high number of apoptotic cell populations as well as caspase activation. Mechanistically, the combination of both compounds upregulated the phosphorylated form of JNK, and the JNK inhibitor SP600125 efficiently reduced cell death caused by the combination treatment. These findings suggest that sorafenib and luteolin combination has a synergistic effect on killing human HCC cells [27] (Figure 4).

##### SC-2001

SC-2001 is a small molecule that blocks protein–protein interaction of anti-apoptotic Bcl-2 family proteins [28]. In addition, it was previously reported that it is able to increase SHP-1 expression and suppress STAT3 phosphorylation in HCC cell lines [29]. The STAT3 pathway was reported to associate with failure of chemotherapy, as well as selection of angiogenic, invasive, and resistant clones [30,31,32,33]. In accordance with that sorafenib-resistant HCC cell lines (SR-1, SR-2, and Huh-7) displayed higher levels of expression of p-STAT3 than sensitive cells [34]. A study reported that SC-2001 had an additive inhibitory effect on tumor growth when used with sorafenib in-vitro. In addition, this combination overcame sorafenib resistance through up-regulating RFX-1 and SHP-1 leading to tumor suppression and facilitation of STAT3 dephosphorylation. RFX-1 is a transcription factor acting as a positive modulator of SHP-1 expression in breast cancer [35]. Regarding in-vivo results, both agents when used together powerfully reduced tumor growth in sorafenib-resistant HCC cell bearing xenograft models and wild type ones as well [36] (Figure 4).

##### Wogonin

Wogonin (5, 7-dihydroxy 8-methoxy flavone) is a natural compound with reported anti-cancer activity. Wogonin exerts its anti-cancer activity through induction of apoptosis in cancer cells and growth suppression of human cancer xenografts in-vivo [37]. A highly potentiated dose dependent cytotoxicity in LDH assay was observed upon using wogonin with sorafenib in Hep3B HCC cell line. This potentiation was empirically detected in the increased number of apoptotic cell populations and caspase cleavage that was reversed by the pan caspase inhibitor. Additionally, wogonin considerably reduced sorafenib-induced autophagy. This autophagy is one of the mechanisms that provides survival advantage to cancer cells leading to resistance to treatment by sorafenib. Inversely, the inhibition of autophagy increased the induced cell death effect of sorafenib on cancer cells [38,39]. In agreement with these results, several studies have reported the promotion of cell death by sorafenib upon autophagy inhibition when combined with other anti-cancer agents [40,41,42,43,44]. Unfortunately, safety issues were aroused in several of these studies due to undesirable side effects or not reaching the expected outcome metric, such as increasing the overall survival [45], demonstrating the benefit of using a natural compound such as wogonin. However, it requires further inspection in-vivo [46] (Figure 4).

##### 419S1 and 420S1

419S1 and 420S1 are multiple tyrosine kinase inhibitors. A transgenic zebrafish platform was used to compare the therapeutic effects of both agents with sorafenib. Zebrafish is a well-known animal model for studying the molecular pathways of human cancers for instance HCC. This is attributed to the conservation of pathways and genes responsible for the physiological development of liver, as well as pathogenesis of the disease between zebrafish and human [47,48]. This has been proven previously through the high analogy of human and zebrafish tumor profiles in respect of ultrasound bio-microscopy and microarray data comparative analysis [49,50]. Sorafenib, 419S1 and 420S1 exhibited anti-angiogeneic effect. 419S1 showed lower hepatoxicity than 420S1 and sorafenib. Furthermore, the therapeutic index (Lethal Concentration 50/Inhibitory Concentration 50) for 419S1 was much greater than for sorafenib and 420S1. Additionally, these compounds reversed the expression levels of cell-cycle-related genes. Using a patient-derived-xenograft assay, it was found that the effectiveness of 419S1 and 420S1 in preventing liver cancer proliferation is better than that of sorafenib alone [51].

##### Doxorubicin

Doxorubicin is a chemotherapeutic agent with anti-mitotic and cytotoxic properties. It intercalates between the base pairs of DNA leading to formation of complexes. In addition, doxorubicin interferes with the topoisomerase II enzyme through stabilization of DNA-topoisomerase II complex, thus preventing the religation step in the ligation–religation cascade catalyzed by this enzyme. A combination of doxorubicin with sorafenib was inspected in a randomized phase III clinical trial with 365 advanced HCC patients. Patients were categorized into two groups where 180 patients administered 60 mg/m^2^ of doxorubicin every 3 weeks along with 400 mg of oral sorafenib twice every day and 176 patients took sorafenib only. The results were evaluated based on the median overall survival (OS) and progression free survival (PFS). In the patients receiving the combination of both agents, median OS was 9.3 months versus 9.4 month in the sorafenib alone group. Regarding PFS, the combination group showed a median PFS of 4.0 months versus 3.7 months in the other group. Unfortunately, the reported adverse events in the combination group was much higher than the sorafenib alone group where high-grade neutropenia and thrombocytopenia arose in 61 and 29 patients, respectively, versus 1 and 4 patients treated with sorafenib only. Accordingly, this study showed that combination did not demonstrate any improvement of OS or PFS in HCC patients [52].

##### Erlotinib

Erlotinib is an oral small molecule inhibitor of epidermal growth factor receptor (EGFR) [53]. The EGFR pathway has been associated with HCC pathogenesis [54]. EGFR activation plays a role in HCC response to sorafenib, proposing that EGFR inhibition may enhance tumor response [54,55]. Accordingly, sorafenib and erlotinib combination was believed to have synergistic effect on tumor growth since each agent is targeting a different pathway. This was attributed to the promising anti-tumor activity in solid tumor HCC patients in a phase I clinical trial [56]. This point was pursued further in a phase III clinical trial including 720 advanced HCC patients with underlying Child–Pugh class A cirrhosis were randomly allocated, 358 patients to sorafenib with placebo and 362 patients to sorafenib along with erlotinib. The primary metric was the median OS, which was roughly similar in both groups: 9.5 months for sorafenib plus erlotinib group versus 8.5 months for sorafenib plus placebo group. However, the overall response was significantly higher in the combination group (6.6%) relative to the placebo group (3.9%) and the disease control rate was considerably lesser (43.9% vs. 52.5%, respectively). Unluckily, the serious treatment-emergent adverse events (AEs) rates were higher in the combination group (58%) relative to the sorafenib plus placebo group (54.6%). For instance, the rates of diarrhea, anorexia, and rash/desquamation were higher in the sorafenib plus erlotinib group. To conclude, adding erlotinib to sorafenib did not enhance survival in patients with advanced HCC [57].

##### Pravastatin

Pravastatin is one of the statins with reported in-vitro and in-vivo inhibitory effect on HCC tumor growth, and a pro-apoptotic effect on HCC cell lines [58,59]. Increase in HMG-CoA reductase’s concentration and activity are associated with HCC explaining the interest in statins due to their effect on this enzyme [60]. Statins inhibit HMG-CoA reductase leading to reduction in mevalonate levels and its products. These products are then used by the cell for post-translational modifications of several proliferation regulators. In addition, chemoresistance shown by HCC cells is due to the deregulation of cholesterol synthesis in the mitochondria [61]. Pravastatin, in particular, has anti-invasive and anti-metastatic activity through restricting MACC-1 (metastasis-associated in colon cancer 1). Accordingly, pravastatin and sorafenib combination seems to be a promising fin HCC. This shall be attributed to the anti-tumor action on two distinctive pathways employed by sorafenibon on the Ras–Raf–MAPK pathway and pravastatin on MACC-1 [62].

The clinical outcome of pravastatin and sorafenib combination versus sorafenib alone, on 323 Child–Pugh A advanced HCC patients was investigated in the PRODIGE-11 clinical trial. 162 patients were administered sorafenib-pravastatin combination while 161 patients took sorafenib only. OS was the primary endpoint and PFS was the secondary endpoint in a duration of 35 months. There was no difference between the two groups in median OS between two treatments groups (sorafenib-pravastatin: 10.7 months versus sorafenib alone: 10.5 months) and no detectable difference in PFS as well. Diarrhea was the main toxicity and it was more severe in combination group (11%) than the solo treatment group (8.9%). Severe nausea and vomiting were infrequent, and there were no deaths related to toxicity. This clinical trial led to the conclusion that the pravastatin–sorafenib combination did not increase survival in advanced HCC patients [63] (Figure 4).

##### Celastrol

Celastrol is a naturally occurring compound and a main active ingredient of Tripterygium wilfordii. It was reported that it has the ability to enhance the anti-tumor effect of conventional anti-cancer drugs in multiple types of tumors [64]. Autocrine PI3K/AKT and VEGF signaling pathways are highly associated with the acquired sorafenib resistance in HCC cells [65]. Celastrol was reported to suppress the AKT pathway and VEGF autocrine system, thus enhancing the anti-cancer activity of sorafenib. This was confirmed by multiple methodologies where MTT IC50 doses of sorafenib and celastrol when used together on HepG2 and Hepa1-6 cell lines were lower than that achieved by sorafenib alone. In addition, celastrol reversed the compensatory activation of the AKT pathway and the autocrine VEGF induced by sorafenib in Western blot and ELISA, respectively. Additionally, celastrol improved the growth inhibition in the cologenic assay and induction of cancer cells’ apoptosis by sorafenib in-vitro and in-vivo in Hepa1-6 tumor-bearing mice models [66] (Figure 4).

##### PI-103

PI-103 is a PI3K and mTOR small molecule inhibitor. PI3K and mTOR proteins are involved with one of the pathways that lead to cell survival and proliferation. PI-103 and sorafenib has been inspected both in-vivo xenograft HCC model induced by subcutaneous inoculation of Huh7 cells in nude mice and in-vitro on Huh7 cell line. The mice were administered with 20 mg/kg sorafenib per day and 5 mg/kg PI-103 every 4 days. There were three groups: mono drug group (sorafenib only), combination group, and control group. There were major differences between the groups. Sorafenib and PI-103 combination inhibited HCC tumorigenesis more efficiently relative to sorafenib only treatment with regard to tumor size [67]. Regarding in-vitro results combining sorafenib and PI-103 synergistically inhibited Huh7 cell proliferation relative to treatment with sorafenib only. On the molecular level, PI-103 repressed key enzymes phosphorylation, such as S6K, AKT, and mTOR, as shown by Western blot blots. Unlike sorafenib, PI-103 did not inhibit the RAS/RAF/MAPK pathway. However, it stimulated MEK ½ and ERK1/2 phosphorylation. Accordingly, the combination of PI-103 and sorafenib intensely repressed both RAS/RAF/MAPK and PI3K/AKT/mTOR pathways. Additionally, sorafenib-PI-103 combination amplified levels of cleaved PARP (apoptosis marker) with 23% in tumor cells relative to single agent treatment as detected in TUNEL assay. In conclusion, the combination of sorafenib and PI-103 has the lead over mono drug therapy on restricting HCC cell proliferation and tumorigenesis [67] (Figure 4).

#### 4.1.2. Rogerafenib (BAY 73-4506)

Regorafenib, a fluoro-derivative of sorafenib, is a multi-target inhibitor approved for the treatment of multiple types of cancers, such as metastatic colorectal cancer, gastrointestinal tumors, and advanced HCC. It inhibits several kinases simultaneously that are crucial in cancer development, such as RET, KIT, BRAF, RAF-1, and BRAFV600E. In addition, it blocks growth factor receptors involved in angiogenesis (VEGFR-1, VEGFR-2, VEGFR-3, and TIE-2) and others involved in metastasis, such as VEGFR-3, PDGFR, and FGFR. Furthermore, rogerafenib inhibits receptors responsible for tumor immunity (CSF1R) [68]. Regorafenib improved OS and PFS in HCC patients who were previously progressive on sorafenib relative to placebo, as reported by the RESORCE clinical trial. Accordingly, it showed superiority to sorafenib. The FDA approved rogerafenib in 2017 for advanced HCC as a second-line treatment (Figure 3). However, it showed common adverse events, including hypertension, diarrhea, and fatigue [69,70] (Table 1). From a health economics point of view, the reported results are controversial. A clinical study conducted the Functional Assessment of Cancer Therapy—Hepatobiliary Questionnaire, and the total score was in favor with the placebo. Rogerafenib has a statistically significant score of 129.31 relative to 133.17 for the placebo where the lower score signifies worse quality of life (QoL) [71]. It is worth mentioning that the original trial did not take into consideration the patients who could not tolerate the side effects of sorafenib [70]. On the other side, another study anticipated that regorafenib can result in an increase of 0.25 QALYs and 19.76 weeks of life (0.38 life years) [72]; another cost effectiveness study reported that regorafenib provided an increase of 0.18 QALYs at a cost of USD 47,112 for advanced HCC patients [73] (Figure 5).

#### 4.1.3. Cabozantinib (XL184, BMS-907351)

Cabozantinib is an orally active tyrosine kinase inhibitor. It targets multiple kinases involved in tumor progression, such as MET and VEGFR2. Additionally, it inhibits other kinases associated with metastasis and drug resistance, including KIT (stem cell factor), RET, ROS1, MER, KIT, TRKB, FLT3, and TIE [68] (Figure 5). Cabozantinib is approved for various types of solid tumors, including progressive medullary thyroid cancer with metastatsis and renal cell carcinoma [74]. In 2019, the FDA officially approved cabozantinib for HCC patients previously treated with sorafenib [75] (Figure 3). The FDA approval was based on a double-blinded phase III clinical trial (CELESTIAL) on patients who did not show progress to sorafenib treatment. In total, 470 patients were assigned randomly to administer 60 mg cabozantinib once per day and 237 patients took placebo. The primary endpoint was the median overall survival (OS). The first group displayed a median OS of 10.2 months versus 8 months in the second (placebo) group. Patients previously treated with sorafenib demonstrated a slight enhancement from 7.2 months to 11.3 months. There was a significant difference between the two groups of the study, where the cabozantinib group showed a median PFS of 5.2 months versus 1.9 months in the other group [76] (Table 1). Regarding the safety profile, it was in harmony with preceding trials. In addition, grade 3 and 4 adverse events were reported including hand-foot syndrome reaction affecting 17% of the patients, as well as hypertension affecting 16% of the patients. These were the most common among the reported adverse events [68].

#### 4.1.4. Lenvatinib (E7080)

Lenvatinib is a urea derivative tyrosine kinase small molecule inhibitor approved for HCC and differentiated thyroid carcinoma treatment. It selectively inhibits VEGF receptors (VEGFR1 (FLT1), VEGFR2 (KDR), and VEGFR3 (FLT4)). In addition, it blocks fibroblast growth factor (FGF) receptors (FGFR1, 2, 3, and 4), PDGFR, KIT, and RET. It was approved by FDA to be used as a first-line treatment for advanced HCC in August 2018 [77] (Figure 3). This was attributed to a phase III clinical trial (REFLECT) in which lenvatinib proved to be non-inferior to sorafenib. Nearly 1000 patients with unresectable HCC were categorized to two arms. The first arm was received once per day, orally, 12 mg if their body weight >60 kg or 8 mg if their body weight <60 kg. The second arm was the taking of 400 mg of sorafenib twice a day. The primary endpoint of this trial was median OS. Patients in the lenvatinib arm showed a median OS of 13.6 months versus 12.3 months in the sorafenib arm, which fulfilled the criteria for non-inferiority. Additionally, PFS was measured as well, and it was considerably higher in the lenvatinib group (7.3 months) versus sorafenib (3.6 months). Furthermore, overall response rate (ORR) was 18% for lenvatinib arm vs. 6.5% for sorafenib arm. The most frequently reported adverse events were hypertension, diarrhea, and decreased appetite followed by weight loss and fatigue. Additionally, lenvatinib showed statistically significant improvements relative to sorafenib in some quality of life (QoL) assessments as the time to clinically meaningful deterioration in role functioning, diarrhea, pain, body image, and nutrition [78] (Table 1). Another study compared lenvatinib with sorafenib and reported that lenvatinib led to an increase of 0.27 incremental life year and 0.23 QALY improvement [79]. 

Unfortunately, resistance to lenvatinib is evolved and another study investigated the mechanism of resistance. To achieve this aim, the authors executed an artificial lethality screen test with lenvatinib. They used a CRISPR–Cas9 library aiming at the human kinome in human HCC cell lines resistant to lenvatinib. The results showed that, only in the presence of lenvatinib, many guide RNAs targeting EGFR were washed-out. This led to the hypothesis that inhibiting EGFR along with lenvatinib will be lethal. To validate this suggestion, the suppression of EGFR expression was performed using EGFR shRNA in a human HCC cell line resistant to lenvatinib in combination with lenvatinib. This results in inhibition of proliferation. Additionally, most HCC cell lines express high EGFR levels and, in these cell lines, the researchers observed a synergistic effect between lenvatinib and the small-molecule EGFR inhibitors (gefitinib or erlotinib). For instance, the combination of lenvatinib and gefitinib entirely inhibited tumor growth in mouse HCC xenograft models, whereas single agent treatments had minute outcomes. This combination led to decrease in proliferation biomarkers and micro-vessel density and increase in apoptotic markers. Likewise, in patient-derived xenograft HCC models with high EGFR levels, the combination provoked noticeable tumor control and was well tolerated by the models. A series of in-vitro mechanistic studies were conducted and revealed another dimension to the synergstic effect of the combination. These studies discovered that lenvatinib inhibited the fibroblast growth factor receptor, which led to feedback activation of the EGFR–PAK2–ERK5 cascade. Activation of this cascade confines the sensitivity of tumor cells to lenvatinib. EGFR inhibitors prevents this feedback activation, explaining the additive inhibitory effect of the combinational therapy. Next, the researchers analyzed EGFR levels in 298 HCC cases using tissue microarray. More than 50% had high levels of EGFR and these patients displayed worse survival. According to the promising preclinical results, a clinical study was conducted to evaluate the safety and anti-tumor activity of lenvatinib–gefitinib combination in 12 patients with EGFR high HCC. The selection criteria included patients whose tumors had shown response and progress on lenvatinib treatment. Overall, 4 patients exhibited partial response; 4 patients had stable disease, while the last 4 showed disease progression after 4–8 weeks of lenvatinib-gefitinib treatment. Taken all together, these results indicate that the novel combination of lenvatinib with an EGFR inhibitor is a promising individualized treatment strategy for advanced HCC patients with high levels of EGFR [80].

### 4.2. Multi-Kinase Inhibitors Not Currently Licensed for HCC

#### 4.2.1. Sunitinib (SU11248)

Sunitinib is an oral small molecule inhibitor with oxindol scaffold. It inhibits a number of tyrosine kinases including PDGFRα/β, VEGFR-1, VEGFR-2, RET, c-Kit, and FLT-3 [81,82]. Mechanistically, sunitinib inhibits the phosphorylation of these tyrosine kinase [83]. Sunitinib demonstrated an anti-angiogenetic and anti-cancer effect in mouse xenograft models [82]. However, it did not show any difference in OS relative to sorafenib in a randomized phase 3 clinical trial. In total, 1074 advanced HCC patients were administered 37.5 mg sunitinib per day versus 400 mg of sorafenib twice daily (Table 1). Additionally, neutropenia and thrombocytopenia were observed with sunitinib, which led to early termination and suspension of the clinical trial [84] (Figure 5).

#### 4.2.2. Erlotinib (CP-358774, OSI-774)

Erlotinib is a quinazoline derivative small molecule inhibitor of EGFR through inhibiting its auto-phosphorylation [85]. EGFR autocrine pathway is involved in tumor development and progression activities, including angiogenesis, cell growth, and metastasis. In the HCC context, EGFR activity was found to have a role in sorafenib resistance, suggesting that EGFR inhibition may increase tumor response to sorafenib [55]. Erlotinib exhibited anti-cancer activity in-vitro and in-vivo in tumor xenograft mouse models [85,86]. In total, 10 trials, 9 phase II trials and 1 phase III trial, were conducted to investigate erlotinib. The rate of tumor response was 0% in 4 of the phase II trials, while it showed a rate <10% in 3 of the phase II and phase III trials, and >20% in the last 2 of the phase II trials. Most of the trials reported a median OS between 6.25 and 15.65 months. The most recurrent grade ¾ toxicities were fatigue, diarrhea, increased transamination of alanine and aspartate, and desquamation. Taken all together, erlotinib provides effective and well-tolerated treatment for advanced HCC. However, more clinical trials need to be conducted to evaluate its efficacy and safety as a single agent, or in combination with other drugs for advanced HCC [87] (Figure 5).

#### 4.2.3. Brivanib (BMS-540215)

Brivanib is an inhibitor of VEGFR-2 and FGFR1 with a pyrrolatriazine scaffold [88,89]. It exerts its mechanism of action through competition with ATP for the binding in the ATP-binding domain of these receptors [90]. Brivanib inhibited tumor growth through its anti-angiogenic effect in-vivo using HCC mouse models [90,91,92]. Brivanib was investigated in a randomized, double blind clinical trial in 395 HCC patients previously treated with sorafenib. Patients were randomly allocated (2:1) to orally receive 800 mg of brivanib daily, as well as best supportive care (BSC) versus placebo and BSC. The primary end point was OS. Results showed a median OS of 9.4 months for brivanib and 8.2 months for the placebo group. In addition, the clinical trial was terminated due to high-grade treatment-related adverse events (AEs), including fatigue, hyponatremia, hypertension, and decreased appetite (Table 1). To conclude, brivanib did not considerably enhance OS [93] (Figure 5).

#### 4.2.4. Cediranib (AZD2171)

Cediranib is a small molecule inhibitor of VEGFR family of proteins exerting its utmost selectivity to VEGFR-2. It is an indole–ether quinazoline-based compound. It acts as an ATP-competitor to its binding domain in these proteins. In addition, it inhibits c-Kit and PDGFRβ as well. Cediranib inhibited the formation of new vessels and regressed the existing vasculature to the tumor cells in an athymic mouse xenograft [94]. Cediranib was investigated in a phase II clinical trial (Table 1) on 17 advanced HCC patients. The patients received 30 mg of cediranib orally once per day for 4 weeks in each cycle. The primary endpoint was PFS rate, to be measured after 3 months. Treatment with cediranib led to 3-month-PFS rate of 77% and median PFS corresponding to 5.3 months. Yet, there was high prevalence of grade 3 toxicities, such as hyperbilirubinemia, in 18% of patients, hypertension in 29% of patients, and hyponatremia in 29% of patients [95] (Figure 5).

#### 4.2.5. Linifanib (ABT-869)

Linifanib is a tyrosine kinase inhibitor of VEGF and PDGF receptors through serving as an ATP-competitive on its binding site in these proteins. It showed effectiveness in reducing the tumor volume in several HCC xenograft tumor models. Furthermore, a combination of linifanib with rapamyin lead to an additive inhibitory effect on the volume of the tumor [96,97]. A phase III trial investigated efficacy and safety of linifanib relative to sorafenib in 1035 HCC patients naive to systemic therapy (Table 1). Patients were divided randomly into two groups, where half the patients took 17.5 mg of linifanib once per day and the other half took 400 mg of sorafenib two times daily. OS was the primary end point of the trial. Linifanib had similar OS as sorafenib. Predetermined boundaries of superiority and non-inferiority were not reached for linifanib. The study did not meet the primary end point and was discontinued due to grade 3 and 4 AEs [98] (Figure 5).

#### 4.2.6. Nintedanib (BIBF1120)

Nintedanib is a small molecule inhibitor with an indolinone scaffold. It inhibits 3 different families of tyrosine kinases involved in the angiogenesis process: VEGFR-1, VEGFR-2 and VEGFR-3, FGFR, PDGFR, and the Src kinase family [99,100]. Nintedanib inhibited VEGF-cell proliferation dependent activity of tumor cells in xenograft HCC mouse models [99]. In addition, it is reported that its mechanism of action may be also related to its role in regulating SHP-1 auto-inhibition, leading to an increase in dephosphorylation of STAT 3 in-vivo causing suppression of proliferation of tumor cells [100] (Figure 5). In a randomized multicenter phase II study conducted by Yen, C.J. et al., after administration of 200 mg in both groups, nintedanib (n = 63) or sorafenib (n = 32), the results were as follows: the median CIR TTP was 2.8 vs. 3.7 months and the median OS 10.2 vs. 10.7 months for nintedanib and sorafenib, respectively; fewer grade 3 or higher AEs (56 vs. 84%), serious AEs (46 vs. 56%), and AEs leading to dose reduction (19 vs. 59%) and drug discontinuation (24 vs. 34%) were seen with the two drugs. Nintedanib AEs were vomiting and nausea, whereas sorafenib lead to ALT/AST elevation, diarrhea, rash, and palmar–plantar erythrodysesthesia syndrome. Together, nintedanib proved similar efficacy to sorafenib for CIR TTP and OS in Asian patients with advanced HCC. AEs can be managed (Table 1) [101].

#### 4.2.7. Refametinib (RDEA119/BAY 869766)

Refametinib is an allosteric inhibitor of the MEK1/2 hydrophobic pocket, inhibiting the enzyme in its inactive form. This enzyme is involved in one of the most reported oncogenic pathways: RAS-RAF-MEK-MAPK/ERK. MEK enzyme is a potential molecular target because of its high selectivity to ERK, which plays a crucial role in driving cell proliferation when activated [102]. A study reported that refametinib demonstrated anticancer activity in-vitro in many HCC cell lines, as well as in-vivo in xenograft and allograft HCC mouse models. Additionally, refametinib displayed an additive inhibitory effect with sorafenib in blocking ERK phosphorylation and inhibiting proliferation of cancer cells in Huh-7, MH3924A allograft mouse models, and Hep3B xenograft mouse models [103] (Figure 5). Furthermore, a phase II study was conducted on 95 patients, 70 of them received Refametinib and the eligible patients received the combination with sorafenib. The majority of patients had liver cirrhosis and hepatitis B viral infection. Refametinib showed 44.8% DCT. Median TTP was 122 days while median OS was 290 days. It worth noting that, the best responders to treatment had RAS mutation. The drug-related AEs were diarrhea, rash, AST elevation, vomiting, and nausea, which require dose modifications in all patients. To sum up, Refametinib plus sorafenib showed anti-tumor activity in patients with HCC and was tolerated at reduced doses by most patients (Table 1) [104].

#### 4.2.8. Vatalanib (PTK787/ZK222584)

Vatalanib is a potent small molecule inhibitor of VEGFR tyrosine kinases family through weakening their auto-phosphorylation action. In addition, it inhibits the c-Kit, PDGFRβ, and c-Fms (colony stimulating factor 1 receptor). On the molecular level, vatalanib interrupted the in-vitro formation of new vasculature and blocked the capillary-like sprout growth [105]. Regarding HCC, it decreased the density of the micro-vessels, inhibited the tumor cells proliferation and induced apoptosis in-vitro and in-vivo in mouse models [106,107,108]. In one study, vatalanib was combined with interferon-α/5-fluorouracil (IFN/5-FU). This combination reduced Akt/ERK/p38MAPK phosphorylation and VEGFR-2 expression [108]. Additionally, a phase I/II clinical trial comprised of 27 patients has been conducted to investigate combining vatalanib with IV doxorubicin for Child–Pugh B cirrhotic HCC patients, where 63% were chronic hepatitis B carriers (Table 1). Patients were administered 750 mg of vatalanib daily along with doxorubicin. Promising results were achieved where the overall response rate was 26.0% and 20% attained stable disease for at least 12 weeks. The OS was 7.3 months and the median PFS was 5.4 months [109] (Figure 5).

#### 4.2.9. Vandetanib (ZD6474)

Vandetanib is a VEGFR-2 and EGFR auto-phosphorylation inhibitor. It has a 4-anilinoquinazoline with basic groups at Carbon 7 of the quinazoline moiety [110,111]. EGFR inhibition lead to its ability to suppress in-vitro HCC cell lines’ processes of adhesion followed by proliferation, migration, and then invasiveness [112]. In-vivo studies reported a positive anti-cancer activity of vandetanib in two types of mouse models: orthotopic and subcutaneous HCC nude models [113] (Figure 5).

#### 4.2.10. Pazopanib (GW786034)

Pazopanib is an indazolypyrimidine based small molecule inhibitor that targets many tyrosine kinase proteins. Its targets are VEGFR-1, -2, and -3; PDGFRα/β; and c-Kit. Specifically, it inhibits VEGFR-2 phosphorylation upon VEGF binding [114]. Pazopanib inhibited the growth of variable tumors in xenograft mice and exhibited anti-angiogenic action in-vivo [115,116]. Regarding HCC, it restricted tumor growth in xenograft HCC mouse models, extending the survival of these models [117]. A phase I clinical trial was conducted to investigate pazopanib as a therapy for advanced HCC in 28 Asian patients. They received dose escalation from 200 mg to 800 mg once per day on multiple 3 weeks cycles (Table 1). Maximum tolerated dose (MTD) was the primary endpoint of the clinical trial. Then, 600 mg was the chosen dose for further improvement of pazopanib in advanced HCC and the safety profile was manageable. Dynamic contrast-enhanced MRI was applied to detect the change in the tumor vasculature. Pazopanib declined tumor vessel leakage, suggesting a direct effect on HCC vasculature as its mechanism for its anti-cancer activity [118] (Figure 5).

#### 4.2.11. Tivantinib (ARQ 197)

Tivantinib is a selective c-Met small molecule inhibitor that exerts its mechanism of action through inhibiting the constitutive and ligand-induced auto-phosphorylation of c-Met, not the conventional ATP-competitive mechanism. Tivantinib induces apoptosis in cancer cell lines, having constitutively activated c-Met, and inhibits tumor growth in xenograft tumor mouse models [119]. Tivantinib-sorafenib combination demonstrated a synergistic cytotoxic activity in HCC cell lines [120]. Consecutively, tivantinib can act on microtubule assembling [121] and glycogen synthase kinase 3 α and β [121,122]. Recently, randomized phase III METIV-HCC was performed on 340 patients with unresectable HCC previously treated with sorafenib to receive placebo or tivantinib 120 mg twice daily (Table 1). Results showed that the median OS among patients treated with tivantinib was 8.4 months compared with 9.1 months in placebo-receiving patients. Moreover, grade 3 or worse AEs (ascites, anemia, abdominal pain, and neutropenia) occurred in 56% compared with 55% of patients who received tivantinib and placebo, respectively. Overall, 22% (50) of patients treated with tivantinib, compared with 16% (18) of patients treated with placebo, died within 30 days of the last dose of the study medication. The most common causes of death were hepatic failure and general deterioration. Thus, tivantinib failed to improve OS as a second line treatment in HCC [123] (Figure 5).

#### 4.2.12. Apatinib (YN968D1)

Apatinib is a potent small molecule inhibitor of phosphorylation of VEGFR-2, c-Kit, PDGFRβ, and c-Src kinase activities [124]. It blocks in-vitro migration, proliferation, and tube formation of HUVEC cell line. In HCC context, it inhibited HCC cell proliferation in-vitro according to CCK-8 assay results. This inhibition was in a dose-dependent manner in six HCC cell lines (SK-Hep-1, HepG2, Hep3B, Huh-7, PLC/PRF/5, and SMMC-7721) and it was correlated to the level of VEGFR-2 expression. In addition, apatinib also prompted the arrest of the cell cycle at G2/M phase and induced HCC apoptosis in-vitro. Moreover, Western blot results confirmed that apatinib treatment at the IC50 concentration inhibited the activation of Akt and ERK1/2 (downstream signal transduction mediators of VEGF/VEGFR-2 pathway) in PLC/PRF/5 cells and SK-Hep-1 cells [125]. Additionally, flow cytometry results revealed that it enhanced apoptosis-related proteins expression levels as cleaved-caspase3 and poly ADP-ribose polymerase (PARP). In harmony with those results, it reduced the expression of anti-apoptotic protein Bcl-2 and upregulated pro-apoptotic protein Bax expression level. These results were in agreement with the observed apatinib-induced HCC apoptosis in-vitro through the mitochondrial-dependent pathway [125]. The anti-cancer activity of apatinib was associated with a reduced density of tumor micro-vessels and an increased median survival in human xenograft mouse models. Finally, a phase II clinical study of apatinib in advanced HCC was initiated. Results proposed that it can be used as a monotherapy in advanced HCC since it is safe and effective [125] (Figure 5). Recently, a single arm phase II clinical trial of apatinib showed that the overall ORR and DCR were 30.4% and 65.2%, respectively. The median OS and PFS were 13.8 and 8.7 months, respectively. The most common drug-related AEs were proteinuria, hypertension, and hand–foot–skin reaction. Collectively, using apatinib showed robust clinical activity, good tolerability, and the AEs can be managed in HCC patients (Table 1) [126].

#### 4.2.13. Dasatinib (BMS-354825)

Dasatinib is a tyrosine kinase small molecule inhibitor. Western blot results demonstrated that it inhibits Src kinase, SFK/FAK, and PI3K/PTEN/Akt pathway. However, it has no effect on Ras/Raf/MEK/ERK and JAK/Stat pathways. Dasatinib inhibits the proliferation, adhesion, and metastasis of HCC cells in-vitro. Its IC50 (0.7–14.2 μM) was measured in nine different cell lines via MTT assay [127]. Furthermore, rosuvastatin [128] and irinotecan [129] were found to have synergistic effect on anti-cancer activity of dasatinib. Rosuvastatin and dasatinib blocked phosphorylation of these proteins: FAK/Src, Ras/Raf, STAT-3, and Akt. This led to enhancing apoptosis via upregulating caspase-3 and downregulating survivin. In addition, this combination inhibited HGF, VEGF, and MMP-9 [128]. The synergistic effect of Irinotecan and dasatinib is due to an enhanced apoptosis rate of HCC cells that is accompanied by mitochondrial dysfunction (Table 1). The enhanced anti-tumor efficacy of SN38 could be explained by the additional inhibition of PLK1, which is triggered by dasatinib [129] (Figure 5).

#### 4.2.14. Imatinib

Imatinib is a tyrosine kinase small molecule inhibitor. It inhibited the proliferation, migration, and metastasis of HCC cells in-vitro and demonstrated anti-cancer effect on HCC xenografts in mice in-vivo. In addition, it reduced AKT phosphorylation and upregulated p62 and LC3 levels in HCC cells and xenografts. Scanning confocal microscopy analysis using a mRFP-GFP-LC3 reporter, as well as transmission electron microscopy analysis, revealed that imatinib obstructed autolysosomes formation, thus suppressing the autophagic flux. In accordance with these results, imatinib was found to reverse the sorafenib-induced autophagy. In addition, imatinib–sorafenib combination exerted an additive inhibitory effect in HCC cells relative to monotherapy. Accordingly, it is proposed that imatinib may target HCC through acting as a tyrosine kinase and autophagy inhibitor simultaneously [130] (Figure 5).

#### 4.2.15. Gefitinib (E1203)

Gefitinib is an oral EGFR small molecule inhibitor that has displayed inhibition of HCC cell lines’ growth in-vitro. EGFR signaling pathway triggers c-Met, hepatocyte growth factor receptor (HGFR), and is highly expressed in almost half of HCCs (47%) leading to survival. However, gefitinib alone did not show activity in a single arm phase II clinical trial conducted on unresectable advanced HCC in 2006 (Table 1) [131]. Accordingly, recent trials have been completed with gefitinib along with other tumor inhibitors, such as genstein. This combination was evaluated in vitro on Hep3B HCC cell line. The IC50 of genistein and gefitinib as calculated by CCK-8 assay were found to be 128.078 and 13.657 µM, respectively. In addition, this combination portentously inhibited cell viability, improved cell apoptosis as detected by flow cytometry results, and reduced phosphorylation of EGFR, VEGFR, and PDGFR. According to Western blots results, the genistein–gefitinib combination enhanced cleaved poly ADP-ribose polymerase and cleaved caspase-3 expression. Furthermore, it was reported that they intensely inhibited Akt/Erk/mTOR signaling pathway. To conclude, these results propose that the genistein–gefitinib combination inhibits HCC proliferation and stimulates apoptosis through inhibition of Akt/Erk/mTOR pathway [132] (Figure 5).

#### 4.2.16. Lapatinib

Lapatinib inhibits EGFR and HER-2/NEU simultaneously via ATP competitive inhibition [133] (Figure 5). Accordingly, auto-phosphorylation and downstream signaling are inhibited along with subsequent fMAPK, AKT, and p70S6 kinase (p70S6K) down-regulation, thus impeding tumor growth [134]. Its tolerance was evaluated initially in heavily pretreated patients with different types of solid tumors in preliminary clinical trials. Results reported that it was tolerated well, with rash and diarrhea being the most frequent AEs [133,134,135]. In total, 26 HCC patients were included in a phase II clinical study with a Fleming Scheme to investigate the safety profile, along with its effectiveness. The patients orally received 1500 mg of lapatinib per day in 28-day cycles. Results showed that patients displayed a median OS of 12.6 months and a median PFS of 1.9 months with the same previously reported side effects. Ten patients demonstrated stable disease and their best response was presented in six patients with SD enduring more than 4 months (Table 1) [135]. A study reported a synergistic effect of lapatinib and celastrol in-vitro. This combination produced strong synergy in growth inhibition (MTT assay) and apoptosis (fluorescence microscope with Hoechst 33,258 staining) in HepG2 cell line relative to single-agent treatments. Additionally, celastrol improved lapatinib’s capability of down regulating EGFR protein expression in HepG2 cells (flow cytometry). Accordingly, this synergy could be used as a novel combination regimen in HCC treatment [136].

#### 4.2.17. Linsitinib (OSI-906/DB06075)

Linsitinib is a potent small molecule inhibitor of insulin receptor and its growth factor pathway receptor-1 (IGF-1R) and insulin receptor. Linsitinib effectively inhibited the phosphorylated form of IGF-1R and its downstream signaling proteins, ERK and p70s6k, in-vitro in multiple human tumor cell lines, including HepG2. Simultaneous inhibition of IGF-1R and the insulin receptor provides an edge and superiority for linsitinib over former classes of anti-IGF drugs. This is attributed to the emerging importance of the insulin receptor upon blocking of IGF-1R by other treatments, as monoclonal antibodies, in activation of IGF axis. Accordingly, the ligands of the blocked IGF-1R (IGF-1 and IGF-2) will not be able to bind to insulin receptor [137]. Linsitinib has currently terminated Phase II clinical trials (Table 1) with advanced HCC patients due to safety issues observed [138] (Figure 5).

#### 4.2.18. Orantinib (TSU-68)

Orantinib is an oral small molecule inhibitor inhibiting multiple tyrosine kinase receptors: VEGFR-2, FGFR, and PDGFR in advanced HCC patients (Table 1) [139]. It was reported that it blocked PDGFR-α phosphorylation in WI-38 cell line in-vitro and inhibited the growth of the tumor in-vivo in Xenograft HCC models induced through co-injecting subcutaneously HuH7/WI-38 cells [140]. Accordingly, a phase I/II clinical trial in patients with unresectable or metastatic HCC was conducted. In phase I, the profiles of safety, tolerability, and pharmacokinetics were investigated based on liver function in 12 patients ranging from no cirrhosis to Child–Pugh class B. TSU-68 was not well tolerated at 400 mg twice per day in Child–Pugh B patients. Accordingly, 200 mg twice per day was established as the succeeding phase II dose. In phase II, an additional 23 patients were included in the study so the full number is 35 patients. The safety and efficacy profiles were measured at the predetermined dose in the previous trial. Time to progression (TTP) and OS were measured. The median TTP was 2.1 months, and the median OS was 13.1 months. The most commonly observed AEs were hypoalbuminemia, malaise, edema, abdominal pain, and AST/ALT elevation. Thus, 200 mg twice a day of orantinib exhibited encouraging initial efficacy with a high safety profile in pre-treated HCC patients [139]. However, it did not prolong OS over placebo in a randomized phase III trial called the ORIENTAL study, performed with orantinib in combination with transcatheter arterial chemoembolization in Japanese patients with Barcelona Clinic Liver Cancer stage B [141] (Figure 5).

#### 4.2.19. Axitinib

Axitinib inhibits VEGFR-1, 2, and 3 potently. In a randomized phase II clinical trial investigating axitinib vs. placebo in advanced and metastatic HCC patients who did not respond to first line treatment with sorafenib, it increased PFS and presented 9.7% as an overall response rate. However, it did not show superiority in OS [142]. Another phase II study reported that axitinib as a second line treatment demonstrated a promising response rate along with good tolerability [143]. An additional phase II trial (Table 1) stated that second-line axitinib displayed a moderate effect and tolerable toxicity in advanced HCC patients who failed on sorafenib monotherapy with a median PFS of 2.2 months and OS of 10.1 months [144] (Figure 5). Preclinical studies proposed that axitinib might increase tumor endothelial cells apoptosis after radiotherapy (RT) in-vitro [145]. Some in vivo studies also demonstrated that axitinib may effectively and safely improve tumor control with RT [145,146]. A small cohort study was conducted on patients to determine the safety and maximum tolerated dose (MTD) of axitinib in combination with radiotherapy for advanced HCC. Nine advanced HCC patients were administered axitinib for 8 weeks during and after RT. Axitinib-RT combination for advanced HCC was found to be tolerated with a maximum tolerable dose of 3 mg twice per day [143]. 

#### 4.2.20. Donafenib

Donafenib is a novel small-molecule inhibitor of multiple kinases, such as VEGFR, PDGFR, and several Raf kinases, hence inhibiting proliferation and angiogenesis of cancer cells. It is a sorafenib derivative with a trideuterated N-methyl group to increase the molecular stability for a better pharmacokinetic profile [147]. Donafenib demonstrated good efficacy and safety profile in preclinical studies [147,148]. A multi-center, open label, and randomized phase II-III clinical trial (Table 1), including 668 patients naïve to systemic therapy, assessed the efficacy and safety of donafenib versus sorafenib in treating unresectable or metastatic HCC. Median OS in patients who took donafenib (12.1 months) was considerably longer than sorafenib (10.3 months). Drug-related adverse events occurred in very few patients receiving donafenib (125 patients) relative to sorafenib (165 patients). Accordingly, donafenib proved superiority over sorafenib in enhancing OS and showed favorable safety and tolerability in advanced HCC Chinese patients, presenting promise as a possible first-line monotherapy for these patients [149] (Figure 5).

#### 4.2.21. Anlotinib

Anlotinib is a novel and potent multi-tyrosine kinase small molecule inhibitor. It has a substantial inhibitory effect against VEGFR 1–3, FGF Receptor 1–4, PDGFR α/β, and c-kit [150]. Anlotinib considerably inhibited the proliferation of HCC cells in-vitro and provoked apoptosis by Bcl-2 inhibition and survivin expression. In addition, it stimulated Bax expression via inactivation of Erk and Akt pathways. Following preclinical trials demonstrated that anlotinib restricted HCC progression [151]. Then, a pilot phase II clinical trial was conducted to investigate its clinical effectiveness and safety as a first- and second-line treatment modality for advanced HCC and to detect the predictive biomarkers. Patients were allocated into two groups: 26 patients who were not previously treated with tyrosine kinase inhibitors and 24 patients pretreated with these inhibitors (Table 1). Both groups received 12 mg per day for 3 weeks/cycle. The primary endpoint of the clinical study was PFS rate, TTP was measured as well, and predictive biomarkers were analyzed and measured in the cytoplasm of the patients. The PFS rate was 80.8% in the first group and 72.5% in the second group. Patients with a baseline CXCL1 (C-X-C motif chemokine ligand 1) plasma level <7.6 ng/μL showed a considerably longer median TTP in both groups. The most common high-grade AEs were hypertension, diarrhea, and hand–foot syndrome. Taken all together, anlotinib exhibited promising safety and efficacy as a first- or second-line treatment modality in advanced HCC. The plasma level of CXCL1 might act as predictive marker for anlotinib efficacy [152] (Figure 5).

#### 4.2.22. Dovitinib (TKI258)

Dovitinib is a potent inhibitor of multiple tyrosine kinases receptor families, including (VEGFR-1, 2, 3), (FGFR1, 2, 3), and (PDGFR-β) [153,154]. In preclinical studies, it inhibited xenograft HCC growth in immune-deficient mice and overcame sorafenib resistance [155,156]. For HCC context, a study investigated the molecular and cellular targets of dovitinib in five HCC cell lines, five endothelial cell lines, and an orthotopic mouse model. Results showed considerable repression of tumor growth and pulmonary metastasis in-vivo in the orthotopic HCC model. PDGFR-β was expressed in 2 HCC cell lines and 4 of the endothelial cell lines. These 4 cell lines expressed FGFR-1, and VEGFR-2 as well. Accordingly, dovitinib was found to inhibit proliferation and motility of endothelial cells at a very low concentration (0.04 μmol/L). However, it was not that potent in inhibiting proliferation or motility of HCC cells. The mechanism of dovitinib was investigated through immuno-histochemical analyses, which revealed that it considerably decreased the density of micro-vessels xenograft tumors, inhibited tumor proliferation, and induced apoptosis. Results propose that dovitinib constrains HCC growth and metastasis favorably through an anti-angiogenic mechanism (Figure 5). 

Angiogenic escape from sorafenib may happen due to activation of angiogenesis associated FGFR pathway. Since the main anti-cancer effect of dovitinib was through blocking angiogenesis. Accordingly, a randomized phase II clinical study of Asian-Pacific patients with advanced HCC was conducted. The selection criteria included patients who were ineligible for surgical and/or loco-regional therapies or their disease progressed after these therapies. Overall, 82 patients received 500 mg of dovitinib orally once a day for 5 days and 83 patients took 400 mg of sorafenib twice a day. The primary and key secondary endpoints were OS and time to progression (TTP). Patients in the dovitinib arm showed a median OS = 8.0 months relative to 8.4 months for the sorafenib arm. Both groups exhibited the same TTP = 4.1 months. Common AEs for dovitinib encompassed diarrhea, decreased appetite, pyrexia, nausea, rash, vomiting, and fatigue. It is worth mentioning that subgroup analysis revealed a considerably higher median OS for patients in the dovitinib group. These patients had plasma soluble VEGFR1 and HGF below median levels relative to those who had these markers at or above the median levels. In conclusion, dovitinib was well tolerated but it did not show superiority to sorafenib. Based on these findings, no succeeding phase III studies have been planned (Table 1) [157].

#### 4.2.23. PD0325901

PD0325901 is a novel MEK1 and MEK2 inhibitor through inhibiting the conversion of ERK to its activated (phosphorylated) form [158]. The p42/p44 ERK/MAPK pathway is upregulated in a majority of human HCC [159,160]. Transforming growth factor alpha (TGF-α) is upregulated in most HCCs as well [161,162]. TGF-α signal through EGFR. EGFR then signals through the MAPK pathway. Thus, TGF-α is a potent stimulator of this pathway [163,164]. PD0325901 suppressed MEK activity and tumor growth in-vitro in TAMH (immortalized murine TGF-α transgenic hepatocyte) cells, taken from the livers of TGF-α transgenic mice. Additionally, it considerably decreased MEK activity in-vivo in athymic mice bearing TAMH flank tumors (Table 1). PD0325901 demonstrated analogous inhibitory activity in HepG2 and Hep3B human HCC cell lines in-vitro and in Hep3B flank tumors in-vivo [165] (Figure 5).

#### 4.2.24. R1498

R1498 is a novel oral small molecule inhibitor of aurora kinases and VEGFR2 involved in angiogenesis and mitosis. It exhibited moderate in-vitro growth inhibition on multiple cell lines with IC50 in micromolar range. R1498 anti-tumor efficacy was compared to that of sorafenib in-vivo on a panel of HCC xenograft mouse models. Results reported superior profile of both efficacy and toxicity relative to sorafenib in all the models. R1498 resulted in 80% inhibition of tumor growth and tumor regression in some xenografts (Table 1). Furthermore, it displayed good in-vivo exposure and wide therapeutic windows in the dose range determination and pharmacokinetic studies making it well tolerated as well [166] (Figure 5).

#### 4.2.25. SGX523

SGX523 is a highly selective MET receptor (mesenchymal epithelial transition factor) inhibitor. It acts as an ATP-competitive inhibitor of the MET receptor. MET and its ligand HGF (hepatocyte growth factor) enhance tumor cells proliferation, invasion, and metastasis in HCC [167,168]. HGF overexpression was reported to hasten HCC progression. In accordance with this finding, genomic analysis in genetically engineered mouse models reported that up-regulated HGF is coupled with HBV-positive HCC patients’ poor prognosis. Lately, HGF/c-Met signaling was associated with drug resistance in tumor microenvironment [167,169] via autocrine signaling. The autocrine signaling of HGF induced phosphorylation of c-Met, thus activating MAPK, as well as AKT pathways, creates tumor sensitivity to SGX523. Partial inhibition of tumor growth was presented by SGX523 monotherapy at 60 mg/kg and at 10 mg/kg sorafenib monotherapy on 2 HCC cell lines: HCC2321 and HCC2309. However, of SGX523 (60 mg/kg)-sorafenib (10 mg/kg) combination (Table 1) gave no major progress in efficacy [170] (Figure 5).

#### 4.2.26. PHA665752

PHA665752 is a small molecule inhibitor of c-Met via inhibiting its phosphorylation and downstreaming PI3K/Akt and MAPK/Erk signaling pathways. HGF/c-Met pathway is deeply involved with metastasis via epithelial-to-mesenchymal transition (EMT). Through trans-differentiation, epithelial cells lose cell to cell contact and acquire mesenchymal features, such as motility and invasion [171]. Losing tight junctions induced by E-cadherin through upregulation of E-box repressors, such as Zeb1 and 2, is one of EMT hallmarks. HGF is one of the growth factors and extracellular signals that activate this transition program [172]. Accordingly, inhibition of proliferation and apoptosis was induced in c-Met positive MHCC97-L and MHCC97-H cells by PHA665752. MHCC97-L and MHCC97-H cell lines showed a mesenchymal phenotype with reduced expression levels of E-cadherin and enhanced c-Met, Fibronectin, and Zeb2 expression levels relative to Huh7 and Hep3B cell lines, which show an epithelial phenotype (Table 1). In accordance with these results, PHA665752 considerably inhibited c-Met positive MHCC97-L and MHCC97-H in xenograft models while c-Met negative cell lines, such as Huh7 and Hep3B cells, were not affected in-vitro or in-vivo [173] (Figure 5).

#### 4.2.27. Tepotinib (EMD 1214063)

Tepotinib is a potent and highly selective small molecule MET inhibitor, showed promising activity in advanced HCC with c-Met overexpression (METamp). Preclinical activity was assessed in 37 HCC patient-derived xenografts (PDXs) in nude mice treated with tepotinib. In addition, it demonstrated potential activity in two Phase Ib/II trials (Table 1), which both met their primary endpoints. One trial enrolled Asian patients without prior systemic therapy (first line) and one enrolled US/European patients with prior sorafenib (second line). Outcomes appeared better in patients with METamp. Thus, the authors further investigated the preclinical and clinical activity of tepotinib in HCC, focusing on high-level METamp, which could be an oncogenic driver in this tumor type [174] (Figure 5). 

Clinical activity was evaluated by analyzing patients with METamp or high-level METamp (defined as mean c-Met gene copy number (GCN) ≥5 and ≥10, respectively; by fluorescence in situ hybridization), who received 500 mg of tepotinib in Phase Ib or II of the first line and second line trials (n = 121). Molecular profiling showed high-level METamp in 2 of 37 HCC PDXs: LIM612 (MET GCN 47.1) and LIPF210 (MET GCN 44). Tepotinib prompted major tumor regression in both of these high-level METamp HCC PDX models (mean tumor volume reduction: 97% and 96%, respectively). Across the two trials, 15 patients treated with tepotinib 500 mg had METamp, of whom 5 showed response (one complete response and four partial responses) and 6 had stable disease as best overall response. In total, 4 patients had high-level METamp (mean MET GCN 14.3, 18.1, 30.2, and 36.2), with best overall response of CR in 1 patient, PR in 2 patients, and SD in 1 patient. It was concluded that High-level METamp may be an oncogenic initiator in HCC that sensitizes cancer cells to c-Met inhibition by tepotinib and a better predictive marker to c-Met inhibitors than METamp [174].

#### 4.2.28. BLU-9931

BLU9931 is a prototype irreversible FGFR4 inhibitor (Figure 4). It exhibited outstanding anti-tumor activity in FGF19-expressing HCC cell lines with FGF19 amplification and intact FGFR4, as well as xenograft models with an intact FGFR4/KLB signaling pathway (Table 1) [175]. FGF19 and its receptor FGFR4 has been shown to induce expression of proliferative markers, such as EGR1 and c-FOS [176]. In HCC pathogenesis, if FGF19 is amplified, this leads to FGF19 overexpression in the hepatocyte resulting in the activation of this pathway and turning from intestine-driven endocrine to autocrine hepatocellular signaling control [177]. Many small molecule inhibitors have been developed but they have unrestrained kinome activity, such as LY-2874455 [178] and ponatinib [179]. This group also developed selective FGFR1–3 inhibitors with moderate to weak potency against FGFR4, such as BGJ398 [180] and AZD4547 [181]. The lack of kinome selectivity results in soft-tissue mineralization and hyperphosphatemia [182]. Using structure-based design principles, BLU9931 was designed to target a cysteine residue found near the ATP-binding site in FGFR4 that is unique among FGFR family members, and rare among other kinases. Kinome-wide selectivity of the compound was confirmed by screening BLU9931 at a 3 μmol/L concentration against 456 kinases and kinase mutants relevant to diseases via KINOME scan method [183]. Treatment of Hep3B cell line (FGF19 is amplified) with BLU9931 led to initiation of caspase 3/7 activity, suggesting that it induced apoptosis at low concentrations which led to inhibition of downstream signaling of FGFR4.

In addition, the anti-tumor efficacy of BLU9931 relative to sorafenib was evaluated. 30 mg/kg sorafenib once per day displayed a modest activity on tumor growth and weight loss was observed in mice control group. However, it prohibited this side effect in a dose-dependent pattern. This led to the conclusion that the highest doses of BLU9931 were both efficacious and well tolerated [175]. 

#### 4.2.29. FGF401

FGF401 is a potent, oral, and highly selective FGFR4 kinase inhibitor. It is the first FGFR4 inhibitor to reach clinical trials, and a phase I/II study is ongoing nowadays in HCC and other types of solid tumors. It was discovered through a high throughput proliferation screening of 436 cancer cell lines obtained from the cancer cell line encyclopedia [184]. The high selectivity to FGFR4 is attributed to the formation of a covalent bond between the 2-formyl tetra-hydro-naphthyridine moiety of FGF401 with a cysteine 552 in the kinase domain of FGFR4. This covalent bond led to the formation of a hemi-thioacetal addition product. FGF401 showed anti-tumor activity in HCC cell lines (Table 1) expressing FGF19 and FGFR4 on their surface like Huh7, SNU878, and Hep3B and xenografts [185] (Figure 4).

Additionally, the pharmacokinetic profile of FGF401 was investigated in RH30 tumor-bearing nude mice. The mice received one dose of intravenous 1 mg per kg or oral 3 mg per kg. FGF401 had somehow short half-life of 1.4 h though the clearance in mice is quite low, as well as 21% oral bioavailability. FGF401 induced tumor stasis at a dose of 10 mg per kg twice a day, as well as tumor regression at these doses: 30 and 100 mg per kg twice a day. These doses were safe and well tolerated in terms of increase in body weight increase. Remarkably, FGF401 anti-tumor effect was superior in Huh7 xenografts relative to once per day 30 mg/kg sorafenib. Sorafenib is reported to cause tumor stasis in xenograft models at 60 g/kg daily [185].

**Table 1 molecules-27-05537-t001:** The most recent in-vivo, in-vitro and clinical trials of tyrosine kinase inhibitors. EGFR, Epidermal Growth Factor Receptor; FGFR, Fibroblast Growth Factor Receptor; VEGFR, Vascular Endothelial Growth Factor Receptor; PDGFR, Platelet Derived Growth Factor Receptor; HGFR, Hepatocyte Growth Factor Receptor; IGF, Insulin Growth Factor Receptor. OS, overall survival; TTP, time to progress; TTRP, Time to radiologic progression; ORR, objective response rate; DCR, disease control rate; PFS, progress free survival; DCR, disease control rate; TEAE, Treatment Emergent Adverse Event.

Drug	Target	Study Design	Sample Size	Results/Primary Endpoint	Secondary Endpoints, Efficacy and Safety	Ref.
Sorafenib vs. Placebo(SHARP)	EGFRs, KIT, PDGFRs, and RAF	phase III; First line; Randomized; Multicenter; Double-blind NCT00105443	n = 602	OS Sorafenib: 10.7, Placebo: 7.9 months	TTP (months): Sorafenib: 5.5; Placebo: 2.8 ORR: 2% DCR: 43% TEAEs: 80%; AEs: 52%	[12]
Regorafenib vs. Placebo(RESORCE)	EGFR1–3, PDGFR-β, FGFR1, KIT, RET and B-RAF	Phase III; Second line; Randomized; International; Double-blind NCT01774344	n = 573	OS Regorafenib: 10.6 Placebo: 7.8 months	TTP (months): Regorafenib: 3.2; Placebo: 1.5 PFS (months): Regorafenib: 3.1; Placebo: 1.5, DCR: 65% ORR: 11% TEAEs: 100%; Grade 3/4 TEAEs: 67%; SAEs: 44	[69,70]
Cabozantinib vs. Placebo (CELESTIAL)	VEGFR1–3, MET, RET and KIT	Phase III; Second line; Randomized; Double-blind NCT01908426	n = 707	OS Cabozantinib: 10.2 Placebo: 8 months	PFS (months): Cabozantinib: 5.2; Placebo: 1.9 DCR: 64%, ORR: 4%; Grade 3/4 AEs: 68%; AEs:50%	[76]
Lenvatinib vs. Sorafenib (REFLECT)	EGFR1–3, FGFR1–4, PDGFRα, RET, and KIT	Phase III; First line; Multicenter; Non inferiority; Open label NCT01761266	n = 954	OS Lenvatinib: 13.6 Sorafenib: 12.3 months	TTP (months): Lenvatinib: 8.9; Sorafenib: 3.7 PFS (months): Lenvatinib: 7.4; Sorafenib: 3.7 DCR:75.5% ORR: 24.1 TEAEs: 99%; Grade ≥ 3 TEAEs: 75%; TEAEs: 43%	[78]
Sunitinib vs. Sorafenib	PDGFRα/β, VEGFR-1, VEGFR-2, RET, c-Kit and FLT-3	Open-label, phase III trial NCT00699374	n = 1073	OS 7.9 versus 10.2 months for sunitinib and sorafenib	PFS: Sunitinib; 3.6, sorafenib 3.0 months TTP sunitinib; 4.1, Sorafenib 3.8 months AEs: grade 3/4	[84]
Erlotinib	EGFR	phase II and phase III clinical trials	n = 1020	OS 6.25–15.65 months	DCR: 42.5–79.6% (PFS) of 6.5–9.0 months, AEs: 3/4 grade toxicities (fatigue, diarrhea, rash)	[87]
Brivanib vs. placebo and best supportive care (sorafenib)	VEGFR-2 and FGFR1	multinational, randomized, double-blind, phase III trial	n = 1150	OS 9.9 months for sorafenib and 9.5 months for brivanib	TTP, ORR, and DCR were similar between the study arms. Most frequent grade 3/4 adverse events for sorafenib and brivanib were similar	[93]
Cediranib	VEGFR-2	single-arm phase II study	n = 17	OS 11.7 (7.5–13.6) months	PFS rate of 77% (60%, 99%). Median PFS was 5.3 (3.5, 9.7) months, stable disease (29%), Grade 3 toxicities: hypertension (29%), hyponatremia (29%), hyperbilirubinemia (18%)	[95]
Linifanib vs. sorafenib	VEGF and PDGF	open-label phase III trial NCT01009593	n = 1035	OS 9.1 months on the linifanib arm and 9.8 months on the sorafenib arm	TTP was 5.4 months (linifanib) and 4.0 months on sorafenib. Best response rate was 13.0% on the linifanib arm vs. 6.9% on the sorafenib arm. Grade 3/4 (AEs); serious AEs; and AEs leading to discontinuation, dose interruption, and reduction were more frequent with linifanib	[98]
Nintedanib vs. sorafenib	VEGFR-1, VEGFR-2 and VEGFR-3, FGFR, PDGFR and Src	randomized, multicenter, open-label, phase I/II study	n = 95	For nintedanib and sorafenib, median OS 10.2 vs. 10.7 months	For nintedanib and sorafenib, respectively, the median CIR TTP was 2.8 vs. 3.7 months Nintedanib-treated patients had fewer grade 3 or higher AEs (56 vs. 84%), serious AEs (46 vs. 56%), and AEs leading to dose reduction (19 vs. 59%) and drug discontinuation (24 vs. 34%).	[101]
Refametinib vs. sorafenib	MEK1/2	phase II study NCT01204177	n = 95	OS 290 days (n = 70)	DCR was 44.8% (primary efficacy analysis; n = 58). TTP was 122 days grade 3 AEs	[104]
Vatalanib in combination with doxorubicin	VEGFRs, c-Kit, PDGFRβ and c-Fms	phase I/II study	n = 27	OS: 7.3 months (range, 0.8–23.6 months)	ORR was 26.0% PFS was 5.4 months (range, 0.27–23.6 months) The commonest grade 3 or 4 non-hematological AEs	[109]
Vandetanib vs. placebo	VEGFR-2 and EGFR	a phase II, randomized, double-blind, placebo-controlled study	n = 67	OS improvement was noticed but statistically insignificant	improved PFS and OS after vandetanib treatment were found, they were statistically insignificant but tumor stabilization rate significant	[186]
Pazopanib	VEGFR-1, -2 and -3, PDGFRα/β and c-Kit	phase I dose-escalating study NCT00370513	n = 28		19 patients (73%) had either partial response or stable disease. Diarrhea, skin hypopigmentation, and AST elevation were the most reported AEs	[118]
Tivantinib vs. placebo	c-Met	a phase 3, randomized, placebo-controlled study NCT01755767	n = 340	OS 8·4 months in the tivantinib group and 9·1 months (7·3–10·4) in the placebo group	Grade 3 or worse AEs (ascites, anemia, abdominal pain, and neutropenia) occurred in 56% compared with 55% of patients who received tivantinib and placebo, respectively	[123]
Apatinib	VEGFR-2, c-Kit, PDGFRβ and c-Src	single-arm, open-label phase II clinical trial NCT03046979	n = 23	The median OS 13.8 months	ORR and DCR were 30.4% and 65.2%, respectively. The median PFS: 8.7 months. The most common treatment-related adverse events were proteinuria (39.1%), hypertension (34.8%), and hand-foot-skin reaction (34.8%).	[126]
Imatinib	AKT, p62 and LC3	phase II clinical trial	n = 17		Grade 3/4 AEs. There was no objective response, and 5 (33%) patients had stable disease. Median time to treatment failure was 1.8 months	[187]
Gefitinib	EGFR	single arm phase II study	n = 31	OS 6.5 months	PFS = 2.8 months, Med OS = 6.5 months. Selected grade 3 AEs: neutropenia; rash; diarrhea. There was only 1 grade 4 AE (neutropenia).	[131]
Lapatinib	EGFR and HER-2/NEU	A multi-institutional phase II study	n = 25	OS 12.6 months	Most common toxicities were diarrhea (73%), nausea (54%), and rash (42%). Ten (40%) patients had stable disease. PFS was 1.9 months	[135]
Linsitinib	IGF-1R	Phase II clinical trials		Not completed due to safety issues observed	Not safe	[138]
Orantinib	VEGFR-2, FGFR and PDGFR	a phase I/II clinical trial in patients with unresectable or metastatic HCC NCT00784290	n = 35	OS 13.1 months	TTP was 2.1 months. Common AEs were hypoalbuminemia, diarrhea, anorexia, abdominal pain, malaise, and edema	[139]
Axitinib	VEGFR-1, 2, 3	Multicenter phase II study	n = 45	OS 10.1 months	DCR was 62.2%, and the RR was 6.7%, (PFS): 2.2 months Axitinib has moderate activity and acceptable toxicity for patients with advanced HCC	[144]
Donafenib vs. sorafenib	VEGFR, PDGFR, and Raf	A Randomized, Open-Label, Parallel-Controlled Phase II-III Trial	n = 688	OS was significantly longer with donafenib (12.1) than sorafenib (10.3) months	PFS: 3.7 vs. 3.6 months. The ORR was 4.6% vs. 2.7% and the disease control rate was 30.8% vs. 28.7%. Drug-related grade ≥ 3 AEs occurred less in donafenib	[149]
Anlotinib	VEGFR 1–3, FGF Receptor 1–4, PDGFR α/β, and c-kit	open-label phase II study (ALTER-0802 study)	n = 50		PFS rate was 80.8% and (TTP) was 5.9 months. Cohort 2, the 1 PFS rate and median TTP was 72.5% and 4.6 months. The most common grade 3–5 AEs were hypertension (8%), diarrhea (8%) and hand-foot syndrome (6%).	[152]
Dovitinib vs. sorafenib	VEGFR-1, 2, 3, FGFR1, 2, 3, and PDGFR-β	Randomized, open-label phase II study	n = 165	The median OS was 8.0 (6.6–9.1) months for dovitinib and 8.4 (5.4–11.3) months for sorafenib	The median TTP per investigator assessment was 4.1 (2.8–4.2) months and 4.1 (2.8–4.3) months for dovitinib and sorafenib, respectively.	[157]
Tepotinib	MET	Phase Ib/II trials	n = 121	Tepotinib induced significant tumor regression in 2 high-level *MET* amp HCC PDX models (mean tumor volume reduction: 97% and 96%, respectively).	High-level *MET* amp may be an oncogenic driver in HCC that sensitizes tumors to MET inhibition with tepotinib. Compared with MET overexpression, high-level *MET* amp could be a better predictive biomarker for MET inhibitors in this setting	[174]
Dasatinib combination with irinotecan	Src kinase, SFK/FAK and PI3K/PTEN/Akt	In-vitro study/nine different cell lines		Dasatinib inhibits the proliferation, adhesion, and metastasis of HCC cells in-vitro.	Dasatinib can reinforce the anti-HCC efficacy of irinotecan/SN38 by downregulation of PLK1 synthesis	[129]
PD0325901	MEK1 and MEK2	HepG2 and Hep3B human HCC cell lines in-vitro and in Hep3B flank tumors in-vivo		PD0325901 suppressed MEK activity and tumor growth in-vitro in TAMH cells, taken from the livers of TGF-α transgenic mice.	Additionally, it considerably decreased MEK activity in-vivo in athymic mice bearing TAMH flank tumors.	[165]
R1498 vs. sorafenib	VEGFR2	In-vivo on a panel of GC and HCC xenografts,		R1498 resulted in 80% inhibition of tumor growth and tumor regression in some xenografts.	R1498 anti-tumor efficacy was compared to that of sorafenib in-vivo on a panel of HCC xenograft mouse models. Results reported superior profile of both efficacy and toxicity relative to sorafenib in all the models.	[166]
SGX523	MET	In-vitro on 2 HCC cell lines: HCC2321 and HCC2309.		Partial inhibition of tumor growth was presented by SGX523 monotherapy at 60 mg/kg and at 10 mg/kg sorafenib monotherapy	SGX523 (60 mg/kg)-sorafenib (10 mg/kg) combination gave no major progress in efficacy	[170]
PHA665752	c-Met	MHCC97-L and MHCC97-H in xenograft models and cell lines as Huh7 and Hep3B cells (in-vitro or in-vivo)		Inhibition of proliferation and apoptosis was induced in c-Met positive MHCC97-L and MHCC97-H cells by PHA665752.	In accordance with these results, PHA665752 considerably inhibited c-Met positive MHCC97-L and MHCC97-H in xenograft models while c-Met negative cell lines as Huh7 and Hep3B cells were not affected in-vitro or in-vivo	[173]
BLU9931	FGFR4	Hep3B cell line		initiation of caspase 3/7 activity, apoptosis, and inhibition of downstream signaling of FGFR4.	BLU9931 is efficacious in tumors with an intact FGFR4 signaling pathway that includes FGF19, FGFR4, and KLB. BLU9931 is the first FGFR4-selective molecule for the treatment of patients with HCC with aberrant FGFR4 signaling.	[175]
FGF401	FGFR4	Huh7, SNU878 and Hep3B cell lines and xenografts in-vivo		FGF401 induced tumor stasis at a dose of 10 mg per kg twice a day, as well as tumor regression at these doses: 30 and 100 mg per kg twice a day. These doses were safe and well tolerated.	FGF401 anti-tumor effect was superior in Huh7 xenografts relative to once per day 30 mg/kg sorafenib	[185]

## 5. Inhibitors of Growth Factors Receptors

### 5.1. Galunisertib (LY2157299)

LY2157299 is a selective small molecule inhibitor of serine/threonine kinase of TGF-β1 [188,189,190]. It has a good safety profile in phase I clinical trials and could be well tolerated in HCC patients [191]. The TGF-β signaling pathway has been shown to be active in HCC and contributes to the epithelial mesenchymal transition (EMT) in HCC models [192,193]. In addition, it is reported that HCC patients had high TGFβ1 plasma levels [194]. High plasma levels of alpha-fetoprotein (AFP) levels in HCC patients are believed to be due to the EMT associated with TGF-β [195], as well as the cancer stem cells triggered through TGF-β signaling [196]. High levels of AFP in this group of patients is also associated with poor prognosis [197]. A phase 2 study (NCT01246986) study was conducted to investigate the safety and efficacy of galunisertib in HCC patients who showed progress on sorafenib or were ineligible to receive sorafenib, and to assess the prognostic value of baseline circulating AFP. A total of 149 advanced HCC patients were allocated into one of two cohorts according to baseline serum AFP. The first cohort comprised of 109 patients with AFP levels more than 1.5× of the upper limit of normal (ULN) while the second cohort included 40 patients with AFP levels less than 1.5× ULN. The first cohort displayed a median OS of 7.3 months and 16.8 months in the second cohort. The most common high-grade AEs were neutropenia, fatigue, anemia, bilirubin elevation, decrease in albumin in blood, and embolism. Taken together, galunisertib showed a manageable safety profile in HCC patients. Lower baseline AFP levels were correlated with longer survival. In accordance with previous results, circulating AFP and TGF-β1 baseline levels and changes from baseline serve as prognostic biomarkers of survival [198].

Primary data showed encouraging results regarding overall patient’s survival with glanisertib treatment. Yet, there were no reports about biomarkers to discriminate patients that would show response to this new agent. Therefore, a study aimed to identify valid biomarkers in preclinical HCC models. The response to treatment with glaunisertib and TGF-β1 was investigated through analyzing the transcriptome of metastatic HCC cell lines using next-generation sequencing-based massive analysis of cDNA ends. In total, 78 frozen HCC samples and 26 ex-vivo HCC tissues, co-cultured with galunisertib, were used to validate the identified mRNAs from the analysis. Five mRNAs were reported: Il11, SKIL, SNAI1, PMEPA1 ANGPTL4, and c4orf26. These mRNAs were intensely upregulated by TGF-β1 and downregulated by galunisertib in many HCC cell lines. However, SKIL and PMEPA1 only were interrelated with endogenous TGF-β1 in the 78 HCC samples. SKIL and PMEPA1 were intensely downregulated and interrelated with endogenous TGF-β1 in the ex-vivo samples. The expression of mRNAs SKIL and PMEPA1 mRNA was significantly higher in tumor tissues compared with controls. In addition, SKIL and PMEPA1 mRNA levels increased as the TGF-β1 mRNA concentrations increased in HCC tissues and strongly downregulated by galunisertib. These recognized target genes may act as biomarkers for predicting the response of HCC patients taking galunisertib [199].

Another study investigated the effect of glanisertib on HCC patients’ response to sorafenib. TGF-β1 pathway is associated with sorafenib-induced resistance in HCC cell lines [200]. Galunisertib and sorafenib combination increased growth inhibition and apoptosis in HCC cell lines and ex-vivo tumor samples, emphasizing a crucial role for TGF-β1 inhibition in overcoming sorafenib resistance [201] and enhanced sorafenib-induced apoptosis [200]. The galunisertib–400 mg of sorafenib twice per day combination was tested in 47 Child–Pugh A patients naïve to systemic therapy. In the beginning, 44 patients received 80 mg of glaunisertib and 3 patients orally received 150 mg twice daily for 2 weeks every 28 days to determine the safety. Then, in the expansion group, all patients took 150 mg of galunisertib twice a day. Primary endpoints incorporated TTP, changes in plasma level of AFP and TGF-β1, OS, response rate, and pharmacokinetics. The pharmacokinetics and safety profiles were in accordance with monotherapy of each agent. Patients who were administered 150 mg twice a day exhibited a median TTP of 4.1 months and a median OS of 18.8 months. Only 2 patients showed partial response, stable disease in 21, and progressive disease in 13 patients. TGF-β1 responders (decrease of >20% from baseline) versus non-responders had longer OS (22.8 vs. 12.0 months). To conclude, the galunisertib–sorafenib combination exhibited acceptable safety profile and an elongated OS outcome [202].

### 5.2. Vactosertib (EW-7197/TEW-7197)

Vactosertib is a potent TGF-β1 receptor kinase inhibitor. TGF-β triggers hepatic stellate cells (HSCs), leading to production of chemokines, cytokines, growth factors and an extensive extracellular matrix [203]. Interactions between HSCs and HCC in the tumor microenvironment (TME) have been shown to induce HCC growth and metastasis [204]. Autocrine and paracrine mechanisms are responsible for crosstalk between cancer cells in TME [205,206]. Tissue inhibitors of metalloproteinases-1 (TIMP-1) is identified as one of the secreted proteins by HSCs and a chief mediator of crosstalk between HSCs and HCC cells induced by TGF-β. TIMP is a potent protein that promotes HCC progression and metastasis. Earlier results reported that TIMP-1 interacts with CD63 on cell surfaces and regulates cell proliferation, migration, and survival. TGF-β signaling upregulates TIMP-1 expression [207]. Accordingly, inhibition of TGF-β signaling using vactosertib considerably reduced the progression in a dose-dependent manner, as indicated by bioluminescence signals and intrahepatic metastasis of HCC, indicated by the reduced number of intrahepatic metastatic nodules, in an SK-HEP1-Luciferase orthotopic xenograft mouse model. In addition, vactosertib inhibited TGF-β-stimulated TIMP-1 secretion by HSCs, as well as the TIMP-1-induced proliferation, motility, and survival of HCC cell lines (SK-HEP1, SNU354, and HepG2). Furthermore, vactosertib disturbed TGF-β induced by EMT and Akt signaling, leading to significant reductions in the motility and growth independent of anchorage of two HCC cell lines, SK-HEP1 and HepG2 cells, as indicated in wound-healing and soft agar colony formation assays [208].

## 6. Immunomodulating Small Molecules

### CS2164

CS2164 is a novel potent orally active multi-target small molecule inhibitor that simultaneously inhibits three major pathways in tumorigenesis, including the angiogenesis-related kinases (VEGFR1-3 PDGFRα, and c-Kit), Aurora B (mitosis-related kinase), and CSF-1R (chronic inflammation-related kinase). CS2164 exerted anti-tumor effects in HCC models in syngeneic Balb/c mice established by injecting H22 hepatoma cell line. In addition, it equally controlled peripheral and in-tumor immune cell populations. It caused CD4+ and CD8+ T cells upregulation in the spleen, but downregulated immunosuppressive immune cells as regulatory T cells, myeloid-derived suppressor cells, and TAMs (tumor-associated macrophages) in the spleen and tumor tissues. Accordingly, CS2164 could be used in immunotherapy potentiation in future cancer treatment [209].

## 7. Small Molecules Inhibiting HCC Pathways

HCC has a complex underlying pathogenesis with multiple pathways involved that play crucial roles in proliferation, metabolism, and apoptosis, among others. Overall, 50% of HCC is known to have activated Wnt/β-catenin signaling pathway, 40–60% is reported to have activated PI3K/Akt pathway. Regarding Myc pathway, it is active in 30–60% of HCC, Hedgehog pathway is activated in 50–60% of HCC while the c-Met pathway is in 30–40% of HCC [210].

### 7.1. Wnt/β-Catenin Signaling

The Wnt/β-catenin pathway is responsible for regulation of multiple cellular processes involved in HCC pathogenesis, such as growth, metastasis, differentiation, and apoptosis. Thus, this pathway represents a potential target for novel molecular treatments [210]. Once Wnt is activated, β-catenin enters the nucleus and recruits its co-activators. Together, these co-activators and β-catenin replace Groucho (β-catenin co-repressor) and interact with T cell-specific factor/lymphoid enhancer-binding factor (TCF/LEF), inducing the Wnt downstream target genes transcription. To disable Wnt signaling, TCF/LEF interacts with Groucho, and binds to its DNA binding domains [211,212]. Many coactivators and corepressors were intensively explored. For instance, it is reported that TCF competes with the tumor suppressor SRY-related HMG-box 1 (SOX1) to associate with β-catenin and leads to downregulation of the Wnt/β-catenin pathway [213]. Accordingly, blocking the abnormal activation of Wnt/β-catenin pathway could be a therapeutic strategy. Deviant Wnt signaling is existent in multiple human cancers, yet no drugs have been accepted to target this pathway [214]. 

#### 7.1.1. YC-1

YC-1 was identified by a Wnt-responsive Super-TOPflash (STF) luciferase reporter assay, as a small molecule inhibitor of the Wnt/β-catenin pathway. YC-1 treatment leads to a reduction in Wnt-regulated transcription through EBP1 p42 isoform, thus inhibiting proliferation of tumor cell. ErbB3 binding protein (EBP1) is homologous to the 38-kDa murine protein p38-2G4, which is a cell cycle-related protein. EBP1 has two isoforms generated by alternative splicing: p42 (spliced form) and p48 (complete form). These two isoforms have essentially contrasting functions in human cancers. Although p42 inhibits cell proliferation and stimulates differentiation, p48 endorses cell survival through different binding protein partners and protein modifications in tumor cells [215]. YC-1 improves p42 isoform binding to the β-catenin/TCF complex and decreases the transcriptional activity of the complex [216] (Figure 6).

Figure 6 Schematic representations of emerging small molecule inhibitors of Wnt signaling pathway and STAT3 signaling pathway. This figure was generated by biorender. Wnt, Wingless and Int-1; TCF/LEF, T-cell factor/lymphoid enhancer-binding factor; GSK-3β, glycogen synthase kinase-3 beta; APC, Adenomatous Polyposis Coli; LRP, lipoprotein receptor-related protein; JAK, Janus Kinase.

#### 7.1.2. FH535

FH535 is an artificial small molecule inhibitor of the Wnt/β-catenin pathway. Two studies reported that FH535–sorafenib combination evoked an additive inhibitory effect on HCC and liver cancer stem cell proliferation, partially through interruption of the bioenergetics of HCC through simultaneous disruption of mitochondrial respiration and glycolysis. Another study reported that FH535 restricts tumor growth in xenograft mouse models. In addition, researchers aimed at exploring the underlying mechanism of FH535 and its derivative, FH535-N in modulating the Wnt/β-catenin-dependent autophagic flux in HCC. There is cumulative proof for the interconnection between the Wnt/β-catenin pathway and autophagy in multiple cancers [217,218]. Autophagy inhibits accumulation of non-functional protein aggregates and organelles that are potentially harmful to the cell and might initiate tumor under stress [219,220]. However, overactive autophagy can also support tumor development [38]. Accordingly, targeting autophagy pathways arise as a novel therapeutic strategy for cancer treatment (Figure 4). In HCC, sorafenib was reported to enhance autophagy [39]. The FH535–sorafenib combination, as well as its derivative–sorafenib combination reduced autophagic flux, as indicated by CytoID. Additionally, these combinations had a synergistic effect on HCC cell proliferation of HCC cell lines, such as Huh7 and Hep3B, and induced apoptosis [221]. 

#### 7.1.3. Mangiferin

Mangiferin is a natural compound with glucosylxanthone scaffold. It is found richly in Mangifera indica leaves and bark. It is previously reported in pharmacological studies that mangiferin has an inhibitory effect on multiple types of cancer cell lines [222]. Its anti-cancer effect is mainly attributed to its inhibitory effect on proteins related to inflammation, cell cycle, and oxidative stress [223]. A recent study reported that mangiferin inhibited the active form of β-catenin [224]. The mechanism of inhibiting Wnt signaling by mangiferin was investigated in-vivo via an orthotopic HCC mouse model with cells expressing luciferase and in-vitro via MHCC97L and HLF cell lines. Oral mangiferin suppressed the in-vivo orthotopic tumor growth, as well as dose-dependent restriction of HCC expansion and invasion. Mangiferin exerted its anti-tumor effect through suppressing (Wilms’ tumor 1) WT1-associated with LEF1 in Wnt signaling [223]. Mangiferin was also reported by an additional study to be selectively useful to an HCC population subset who have diethynitrosamine (DEN)-induced HCC. Healthy Sprague Dawley rats drank water containing 0.01% DEN for 12 weeks to induce HCC. This was followed by administrating 50 mg of mangiferin for 8 weeks. Results indicated that it has anti-cancer effect against DEN-induced HCC. These results were validated by estimating the expression of apoptotic proteins along with histological analysis of liver tissue. This is in addition to evaluating the biochemical, oxidative stress markers, and tumor marker level in the DEN+ and DEN- rats liver treated with mangiferin [225] (Figure 6).

#### 7.1.4. IC-2 and PN-3-13

The presence of liver cancer stem cells (CSCs) or CD44+ cells contribute to HCC metastasis, recurrence, and resistance to chemo/radiotherapy [226,227,228]. The WNT signaling pathway is linked to the preservation of CSCs stemness [229]. Genetic and pharmacological inhibition of WNT signaling pathway suppress CSC features as self-renewal, proliferation, and invasion, in numerous kinds of cancer [230,231]. Accordingly, a study investigated the use of small molecule Wnt inhibitors to eliminate liver CSCs (flow cytometry and sphere forming assay). In Huh7 human HCC cells, IC-2 and PN-3-13 suppressed cell viability, WNT signaling activity and dramatically reduced CD44+ and CD44- Huh7 cells’ ability to form spheres. Additionally, IC-2 lead to a reduction in CD133+ HepG2 cells and CD90+ HLF cells subpopulations (express CSC markers). Finally, inhibitory activity of IC-2 on liver CSCs was detected in a xenograft model using CD44+ Huh7 cells. Taken together, IC-2 can serve as a promising therapeutic agent to improve the prognosis of HCC patients [232]. 

### 7.2. RAS-RAF-ERK Signaling

#### Rigosertib (ON-01910)

Rigosertib is a RAS and polo-like kinase 1 (PLK1) signaling inhibitor with benzyl styryl sulfone scaffold. It was reported that upregulated PLK1 is considerably associated with poor patient survival. RAS–RAF–ERK signaling pathway is one of the major pathways involved in advanced HCC and the chief therapeutic agents are tyrosine kinase inhibitors of the proteins in this pathway, indicating its importance. RAS protein has multiple isoforms which were not reported to have a vital role in developing HCC. Still, numerous studies imply that HRAS isoform can serve as potent oncogene in HCC, but its inhibition with small molecule inhibitors was not done yet. Furthermore, the PLK1 is reported to be stimulated by RAS–RAF pathway. Rigosertib, being able to inhibit simultaneously RAS and PLK-1, was investigated in advanced HCC patients who have upregulated PLK1 and RAS isoform, HRAS levels. A study aimed at analyzing the effect of rigosertib on PLK1 and HRAS expression in HCC. It decreased in-vitro cell proliferation and led to cell cycle arrest in human HCC cell lines. Additionally, it greatly suppressed activation of ERK and AKT proteins in HCC cells. According to HCC patients’ data analysis, the cell cycle promoting PLK1 is upregulated during HCC development, while HRAS is upregulated in advanced HCC. Simultaneous upregulation of PLK1 and HRAS showed collective effects on patient outcome. This emphasizes the prominence of these genes and their accompanying pathways in HCC. That study newly revealed the therapeutic capability of rigosertib in HCC by inhibition via simultaneous inhibition of PLK1 activation and major RAS-pathways, unveiling a novel therapeutic method for HCC [233].

### 7.3. JAK/STAT3 Signaling

#### 7.3.1. 2-Ethoxystypandrone

2-ethoxystypandrone is a natural and novel compound that targets STAT3 signaling pathway. It inhibits STAT3 potently with a reported IC50 of 7.75 ± 0.18 μM. STAT3 is known to be an oncogene that is constitutively triggered in HCC cells and HCC CSCs. Activated STAT3 exerts a key role in holding cancer stemness property of HCC CSCs, which is responsible for HCC initiation, metastasis, and relapse along with drug resistance [234]. A study conducted a STAT3-dependent luciferase reporter gene assay and results revealed that 2-ethoxystypandrone blocked IL-6-induced and constitutive activation of STAT3 phosphorylation in HCC. In addition, it inhibited HCC cells survival in-vitro in MTT assay and obstructed the formation of tumor spheres in tumor sphere formation assay. These results suggest that it inhibits HCC CSCs self-renewal capacity. Finally, it prompted apoptosis of HCC CSCs in a dose-dependent pattern as indicated by flow cytometry results [234] (Figure 6).

#### 7.3.2. FLLL32

FLLL32 is a novel curcumin analog that was designed to selectively target STAT3. It was designed to better interact with binding sites of JAK2 and dimerization domain of STAT3. The mechanism of the inhibitory effect of IL-6-induced STAT3 activation is that FLLL32 blocked JAK2 mediated STAT3 phosphorylation and dimerization. It inhibited IL-6-induced STAT3 in a dose-dependent manner in Hep3B and SNU-398 cell lines as indicated by Western blot. Moreover, FLL32 treatment reversed the IL-6-induced STAT3 nuclear translocation. FLL32 is said to be selective to STAT3 since it had no effect on IFN-γ STAT1 induced phosphorylation [235] (Figure 6).

#### 7.3.3. XZH-5

XZH-5 is a novel and selective small molecule inhibitor of STAT3. XZH-5 binds to STAT3 SH2 domain where it forms 4 hydrogen bonds with the SH2 domain: two with Arg609, one with Ser636 and one with Lys591. In addition, the tri-flurobenzyl ring had a hydrophobic interaction with a side pocket of SH2 domain. XZH-5 reduced constitutive STAT3 phosphorylation at Tyr705 in HepG2, Huh-7, SNU-387, and SNU-398 cells as indicated by Western blot and reduced the expression of STAT3 downstream genes as indicated by RT-PCR. The inhibition of STAT3 in HCC cells resulted in the induction of apoptosis and reduction in colony forming ability [236]. In addition, XZH-5 also inhibited IL-6-induced STAT3 phosphorylation, nuclear translocation and STAT3 DNA binding activity. However, it had no effect on IFN-γ-induced STAT1 phosphorylation, indicating the more selective effects on STAT3. These results suggested that XZH-5 may serve as a lead compound for development of selective STAT3 small molecule inhibitors for HCC therapy [236] (Figure 6).

### 7.4. PI3K/Akt/mTOR Pathway

#### SC66

SC66 is a novel allosteric AKT small molecule inhibitor. It inhibits AKT through 2 separate mechanisms of actions: enabling its ubiquitination and blocking pleckstrin homology (PH) domain binding to PIP3 [237]. SC66 was tested in-vitro in multiple cell lines and results revealed that it decreased cell viability in a dose- and time-dependent pattern. In addition, it reduced colony formation capacity in Hep3B, HA22T/VGH, HepG2, Huh7, and PLC/PRF/5 cell lines. Increase in caspase activity confirmed the effect of SC66 in stimulating apoptosis in Hep3B and Huh7 cell lines. Additionally, it decreased the total and phosphorylated forms of AKT and, thus, decreased phosphorylation levels of mTOR, the downstream target protein of phosphorylated AKT, as shown by Western blot results. Modifications in cytoskeleton organization was observed along with a reduction in E-cadherin expression confirmed by the up-regulation of Snail protein levels. Snail protein negatively regulates transcription factor of E-cadherin gene [238]. Anoikis, which is defined as programmed cell death from the loss of cell adhesion to the extracellular matrix, was observed in Hep3B cells. SC66 in combination with chemotherapy (doxorubicin) and targeted agents (everolimus) had an additive inhibitory anti-tumor effect. Finally, SC66 inhibited tumor growth in-vivo in xenograft models, with an analogous mechanism detected in the in-vitro model [239].

## 8. Small Molecules Targeting Various Molecular Targets

### 8.1. CMO

CMO (2-(3-chlorobenzo[b]thiophen-2-yl)-5-(3-methoxyphenyl)-1, 3, 4-oxadiazole) has been identified as a lead compound by molecular docking analysis. CMO interacts with the hydrophobic region of p65 protein thus causing inhibition of nuclear factor-kappa-B transcription complex (NF-κB) and its signaling pathway [240]. Constitutive activation of NF-κB is linked with the progression of human malignancies, including HCC [241,242]. CMO induced dose-and time-dependent anti-proliferative effect against HepG2 and HCCLM3 cells [243]. Flow cytometric analysis showed that CMO significantly increased the percentage of sub-G1 (hypodiploid) cell population and induced apoptosis in HCCLM3 and HepG2 cell lines [240]. Caspase-activated DNase-mediated fragmentation of the genomic DNA is a remarkable event in the cells committed to undergo apoptosis, which results in the formation of cells with lesser DNA content known as hypodiploid cells [244,245]. CMO was found to decrease the phosphorylation of IκB (Ser 32) in the cytoplasmic extract of HepG2 cells [240]. Phosphorylation and proteolytic degradation of IκB is essential for posttranslational activation of NF-κB [246]. Next, the study investigated whether the knockdown of p65 using siRNA could considerably block the increase in CMO induced caspase-3/7 activation in HepG2 cells. In cells transfected with control siRNA, CMO significantly increased caspase 3/7 activation, thus inducing apoptosis [240] (Table 2).

### 8.2. APG-1387 (Apoptosis Inhibitor)

APG-1387 is a novel small molecule apoptosis inhibitor targeting IAPs (inhibitor of apoptosis proteins) via acting as a mimetic of SMAC (second mitochondria-derived activator of caspases). IAPs play a key role in multiple processes of tumor progression, including cell survival, resistance to chemotherapy, along with poor prognosis. In addition, cellular IAPs affect HBV clearance through suppressing a TNF pathway. SMAC mimetics are known to cause a rapid drop in serum DNA of HBV, as well as its surface and core antigens in animal models [247]. Accordingly, they would have a great potential in treating HBV+ HCC patients. Moreover, IAPs, through their ubiquitin-E3 ligases, play a crucial role in immune regulation where they control innate immune signaling through triggering NF-κB signaling pathway [248,249,250]. APG-1387 increased natural killer cell counts by 5-fold in a xenograft model relative to control group. In-vitro, APG-1387 decreased regulatory T cells differentiation thus positively modulated T cells. Additionally, it down-regulated programmed cell death-1 (PD1) expression in CD4+ T cell. However, APG-1387 did not affect memory T cells. Therefore, it is suggested that APG1387 might be a potential candidate for combinational therapy with monoclonal antibodies against PD-1 antibody treatment to overcome weak response of HBV+ HCC patients to checkpoint inhibitors [251] (Table 2).

### 8.3. AC-73 (AN-465/42834501) (CD147)

AC-73 is a CD147 (a type I transmembrane glycoprotein) prototype inhibitor. CD147 is upregulated in numerous cancers and plays crucial roles in tumor progression. This is attributed to its role in stimulating HCC motility and invasion. AC-73 was discovered through virtual screening of over 300,000 compounds using a pharmacophore model. AC-73 can specifically disrupt CD147 dimerization through binding to two amino acids in the N-terminal IgC2 domain. These two amino acids are found in the dimer interface of CD147, as proposed by molecular docking and mutagenesis experiments [252]. Furthermore, AC-73 suppressed HCC metastasis by inhibiting matrix metalloproteinase via reducing CD147/ERK1/2/STAT3 signaling pathway. In agreement with that, AC-73 decreased progression of the disease in an orthotopic mouse model of HCC metastasis [252] (Table 2).

### 8.4. VO-OHpic (PTEN Inhibitor)

VO-OHpic is a small molecule inhibitor of the tumor suppressor gene phosphatase and tensin homolog (PTEN). Although PTEN mutations seldom happen in HCC, heterozygosity of PTEN is detected in 32–44% of HCC patients leading to decrease in PTEN expression. VO-OHpic suppressed cell proliferation, cell viability, and colony formation. In addition, it triggered senescence in b-galactosidase enzyme activity in cell lines with low PTEN expression (Hep3B) and to a lesser degree in cell lines with high PTEN expression (PLC/PRF/5). However, it did not affect PTEN-negative cell lines (SNU475). In addition, VO-OHpic has an additive inhibitory effect with PI3K/mTOR and RAF/MEK/ERK pathway inhibitors on cell viability of Hep3B cells only. In addition, it considerably inhibited tumor growth in Hep3B nude mice bearing xenografts. Accordingly, inhibition of PTEN may be a potential therapeutic approach for HCC subpopulation who have low PTEN expression [253] (Table 2).

### 8.5. Rubone (miR34a)

Rubone is a lead candidate as a small molecule activator of miR34a recognized via its capability to selectively increase the expression miR34a in HCC cells in HCC cell-based miR34a luciferase reporter system. miR34a act as a tumor suppressor and it is reported that it is downregulated or silenced in many cancers, such as HCC. Rubone upregulated miR34a in wild type or mutated p53 HCC cells but not in cells carrying p53 deletions. Remarkably, rubone did not have any effect on growth of non-tumorigenic human liver cells. In a mouse xenograft model of HCC, rubone considerably inhibited HCC tumor growth, displaying superiority relative to sorafenib in-vitro and in-vivo. Mechanistically, rubone downregulated cyclin D1, Bcl-2, and miR34a target genes. Additionally, it improved p53 occupancy on the miR34a promoter. Taken together, these findings present preclinical evidence that rubone is a lead compound of a new class of HCC therapy based on its ability to restore miR34a function as a tumor suppressor [254] (Table 2).

### 8.6. FQI1

Factor quinolinone inhibitor 1 is a small molecule inhibitor of the transcription factor LSF. LSF is upregulated in HCC patient samples and cell lines. In addition, it enhances oncogenesis in xenograft rodent models of HCC proposing that it is a promising protein target for chemotherapy. FQI1 blocks LSF DNA-binding activity in-vitro and eliminates transcriptional stimulation of LSF-dependent reporter constructs. Moreover, FQI1 exhibits anti-proliferative property in many cell lines. Furthermore, FQI1 dramatically inhibited HCC growth in a mouse xenograft model with no observable general tissue cytotoxicity [255] (Table 2).

### 8.7. AUY922 (Luminespib)

Luminespib (resorcinylic isoxazole amide) is a potent third generation small molecule inhibitor of heat shock protein 90 (HSP90). HSP-90 is one of the most significant molecular chaperones and many protein partners interacting with HSP-90 have been identified. It modulates oncoproteins positively via promoting their stability, function, and activity, therefore participating in the malignant phenotype. Some of the oncogenic protein partners of HSP90 are involved in pathways of cell growth and proliferation (such as EGFR and β-catenin), apoptosis signaling pathways (such as p53 and AKT), as well as angiogenesis (such as VEGFR). Most of these proteins are frequently dysregulated in HCC. It was reported that HSP90 expression profile was considerably greater in HCC tissues than in cirrhotic peri-tumoral liver tissues. Luminespib decreased HCC cells proliferation and viability in a dose-dependent pattern. However, it did not affect normal hepatocytes. In addition, AUY922 repressed tumor growth in-vivo in a xenograft model [256] (Table 2).

### 8.8. Compound 81 (CXCR6)

Compound 81 is a potent, orally active, and selective antagonist to human CXCR6 receptor signaling pathway with an EC50 of 40 nM. The chemokine system has a crucial role in facilitating a pro-inflammatory TME thus promoting HCC growth. The CXCR6 receptor and its ligand CXCL16 are upregulated in HCC cell lines and tumor tissues. CXCR6 expression is correlated with increase in neutrophils in HCC tissues leading to poor prognosis. Compound 81 significantly decreased tumor growth in mouse xenograft HCC model [257] (Table 2).

### 8.9. Cambinol (SIRT-1)

Cambinol is a small molecule inhibitor of SIRT-1 member of Sirtuins. It a highly conserved family of NAD+-dependent enzymes that is comprised of 7 members that control histone and non-histone regulatory proteins’ activity. SIRT1 promotes longevity and suppresses the initiation of some cancers. However, SIRT1 is reported to have contradictory functions acting as a tumor suppressor, as well as an oncogenic protein. SIRT1 is strongly and consistently upregulated in multiple HCC cell lines (Hep3B, HuH7, HepKK1, skHep1, HepG2). SIRT1 inhibition impaired proliferation of HCC cells in-vitro and cambinol treatment resulted in an overall lower tumor burden in-vivo in an orthotopic xenograft model [258] (Table 2).

### 8.10. BI 2536

BI 2536 is a small molecule inhibitor of Polo-like kinase 1 (Plk1). Analysis of Plk1 expression in samples of HCC patients assured that Plk1 expression levels are higher in tumor cells than the normal liver tissue. To investigate whether Plk1 is a suitable target in HCC, its therapeutic effectiveness of Plk1 inhibition was compared in 3 different models: HCC cell lines, nude mice xenografted with HCC, and in a transgenic mouse model (TGF α/c-myc) evolving HCC endogenously [259]. BI 2536 decreased the viability of HCC cell lines, suppressed HCC xenograft progression in nude mice and lessened the number of dysplastic foci as indicated by the low number of Ki-67+ cells inside the foci, indicating weakened tumorigenesis. However, it had no significant effect on HCC progression in the transgenic mouse model due to low intra-tumoral levels of Plk1, which is a resistance mechanism to Plk1 inhibitor. Accordingly, Plk1 inhibitors reaching adequate intra-tumoral levels are highly promising in HCC treatment [259] (Table 2).

### 8.11. THZ1

THZ1 is a phenylamino-pyrimidine small molecule inhibitor of cyclin dependent kinase 7 (CDK7) at the nM range. It interacts with cyst 312 residue with an irreversible covalent bond, thus obstructing the ATP cleft in the kinase domain and permanently inhibiting CDK7 kinase activity. CDK7 is one of the most important components of the trans-acting super enhancer (SE) complex along with other proteins that were found to be commonly overexpressed in human HCCs and were associated with the poor prognosis of patients. One of mechanism of molecular pathogenesis of HCC is acquiring SEs at multiple noticeable oncogenes to drive their vital expression [260]. THZ1 treatment efficiently suppressed the serine residues phosphorylation at different locations (2, 5, and 7) of the polymerase II C-terminal domain in HepG2 and Huh-7 cells, in addition to its growth inhibitory effect. Thus, it is postulated that THZ1 treatment may inactivate the transcription initiation and elongation of SE-associated genes, which are highly dependent on CDK7 activity. These target genes which respond to THZ1 were considerably augmented in biological processes, including transcription, metabolic processes, and gene expression, cell cycle progress, and DNA repair as reported by Gene Ontology analysis. These findings propose that CDK7 inhibition signifies an effective way to suppress the expression of multiple HCC-SE genes involved with HCC cells’ viability at the same time. Moreover, THZ1 considerably decreased the liver tumor size, representing an anti-cancer activity of THZ1 in-vivo [260] (Table 2).

### 8.12. IPA-3

IPA-3 is a highly selective, non-ATP-competitive small molecule allosteric inhibitor of p21-activated kinase 1 (PAK1). Increase in PAK1 activity is associated with tumorigenesis [261]. The inhibitory action of IPA-3 is partially executed by forming a covalent bond with PAK1 regulatory domain, which, in turn, prevented the interaction with Cdc42 or other GTPase enhancers [262]. Targeting PAK1 regulatory domain provides IPA-3 with a selectivity advantage since this domain is less conserved within kinases [261]. Previous reports demonstrated that PAK1 triggers the subcellular translocation and, thus, the activity of nuclear factor light-chain enhancer of activated B cells (NF-ĸB), and promotes cell survival [263,264].

IPA-3 was investigated in-vitro on 5 different cell lines. MTT assay was conducted on HepG2 and H2P (primary), MHCC97L and H2M (metastatic), and non-tumorigenic MIHA (immortalized liver). Results showed that it inhibited the proliferation of H2M cells in a time-dependent and dose-dependent pattern. It reduced the number of metastatic HCC cells more than the primary HCC cells. However, MIHA had the highest resistance to IPA-3, proposing that IPA-3 inhibit hepatoma cell proliferation with a negligible activity on normal liver cells. Mechanistically, IPA-3 led to a decrease in cell cycle entry, as indicated by the BrdU labeling assay. Notably, IPA-3 only had a marginal impact on the incorporation rate of BrdU on the primary cell lines, suggesting that it is highly selective for the metastatic HCC cell lines. These findings were in agreement with a colony formation assay. Taking all the results together, IPA-3 inhibited HCC cell proliferation in an ascending order of non-transformed hepatocytes < primary HCC < metastatic HCC [265] (Table 1). Additionally, Western blotting analysis demonstrated that IPA-3 reduced the phosphorylation PAK1, as well as phosphorylated JNK kinase (the downstream target of PAK1) in a dose-dependent manner. The mechanisms through which IPA-3 exerts its anti-cancer activity via inducing apoptosis and blocking activation process of NF-ĸB. These results were obtained in H2M cell line by giving a positive signal of annexin V-7ADD staining. Finally, preclinical studies on nude mouse xenograft proved that it decreased the tumor growth rate and volume, suggesting that IPA-3 is effective in-vivo as well [265] (Table 1).

### 8.13. Alisertib (AURKA Inhibitor)

Alisertib is a potent small molecule AURKA inhibitor via inhibiting its auto-phosphorylation at Thr288. It was discovered in a study that was conducted to discover new HCC therapeutic targets through previously reported genomic data collected from 1061 HCC patients and carrying out integrative analysis to identify considerably mutated genes. The analysis revealed 7 mutated genes. Some of these genes were associated with classical driver genes mutations as TP53 and TERT. The results also identified TERT focal amplifications and other potential drug targets including AURKA. Alisertib intensely suppressed cell viability and colony-forming ability of numerous HCC cell lines in-vitro as Hep3B, HepG2, SNU449, and SNU182. Furthermore, it strongly restricted SNU449 and SNU182 cell lines migration [266] (Table 2).

## 9. Discussion

HCC is still regarded as a challenge by healthcare professionals. Even with medical improvements, the prognosis of patients with advanced disease is poor due to the underlying complexity of the pathogenesis of the disease and involvement of various molecular targets. However, recent trials may transfigure the treatment protocol. After 10 years or more of a relative lack of progress in HCC treatments, speedy changes have happened recently. Numerous new molecules have been accepted, and others are currently under investigation. In addition, new molecular targets are discovered in line with the remarkable developments in tumor cell genomes, transcriptomics, epigenomics, and proteomics. Thus, gates are being opened to new combinational strategies to improve the prognosis of the disease and target multiple crucial molecular targets simultaneously.

Combination treatments display several advantages because this strategy targets important pathways in a manner that is typically synergistic or additive. The combination of anti-cancer medications improves efficacy in comparison to the monotherapy by halting mitotically active cells, lowering cancer stem cell numbers, and triggering apoptosis [267]. In addition, this strategy may concurrently diminish drug resistance [267,268], while offering therapeutic anti-cancer advantages. Another advantage of a combinational approach is mitigating the clonal heterogeneity associated with improved response rates [269]. Finally, drug combinations allow for the use of separate medications in lower dosages while maintaining therapeutic efficacy reducing the toxicity of the treatment [268]. In this manner, a combination treatment of Atezolizumab and Bevacizumab became the first-line treatment for advanced HCC, receiving European Medicines Agency (EMA) approval in late 2020 replacing all monotherapy options, including sorafenib [270]. However, there is not a treatment plan that fits all with a significant number of HCC patients did not respond to this combination plan and, thus, other combination therapies are still needed.

Despite their benefits, developing a new combination therapy for cancer is expensive, takes a long time and comes with a number of difficulties. The possible drug interactions and pharmacokinetics of co-administered medications that may affect the regimen’s therapeutic efficacy are a difficult component of combination therapy [271,272]. Additionally, it may be required to administer sub-optimal amounts of the combined medications to prevent toxicity [273]. It is worth noting that the majority of medication combinations have traditionally been developed in empirical settings. In such a setting, a thorough analysis of the mechanism of actions is infrequently carried out for the prediction of efficient combinations [269]. Therefore, new tactics for developing combination therapies that deliver good outcomes at a reasonable cost are being considered. Several studies reported revolutionary approaches in this regard, and applied them in solid tumors, including HCC, such as cancer drug atlas [274] and CombiPlex platform [272]. These rational methods appear promising to identify novel and unexpected combination therapies and, even more, identifying personalized multidrug therapies, thus giving new hope to advanced HCC patients. 

## Figures and Tables

**Figure 1 molecules-27-05537-f001:**
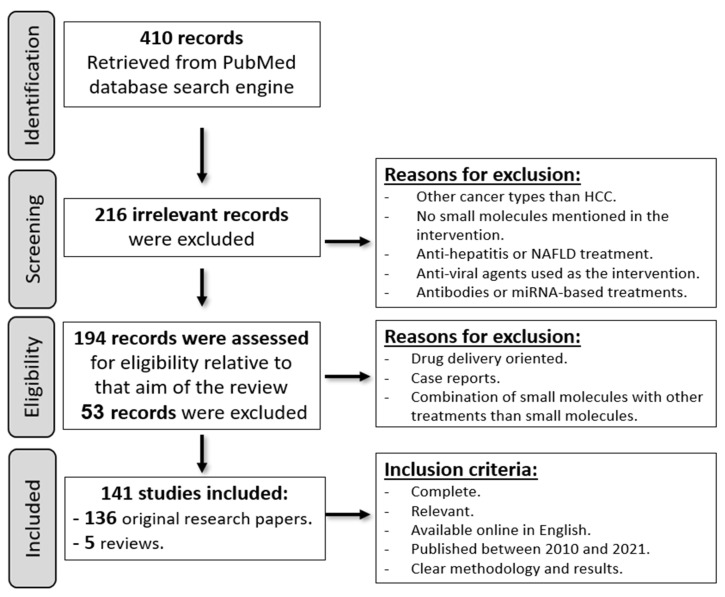
Flow diagram of the literature screening process including number of records retrieved from search databases, exclusion and inclusion criteria.

**Figure 2 molecules-27-05537-f002:**
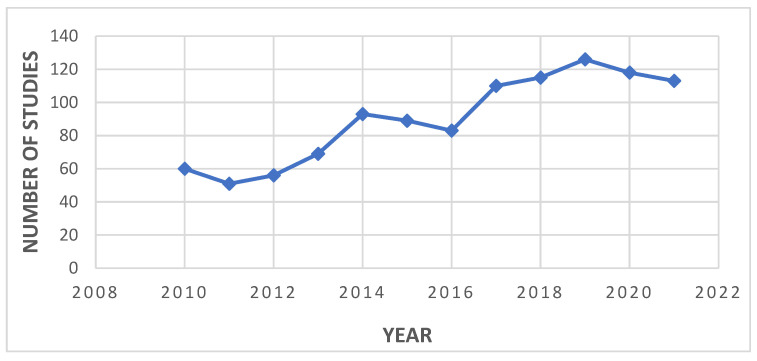
Schematic representation of research interest toward small molecule inhibitors in liver cancer treatment. *Y*-axis represents number of studies and *X*-axis represent years. These data were collected from PubMed.

**Figure 3 molecules-27-05537-f003:**
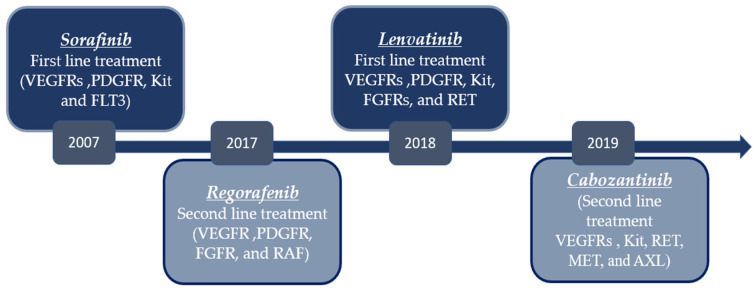
The chronological sequence of the currently FDA approved small molecule inhibitors in HCC and their primary molecular targets.

**Figure 4 molecules-27-05537-f004:**
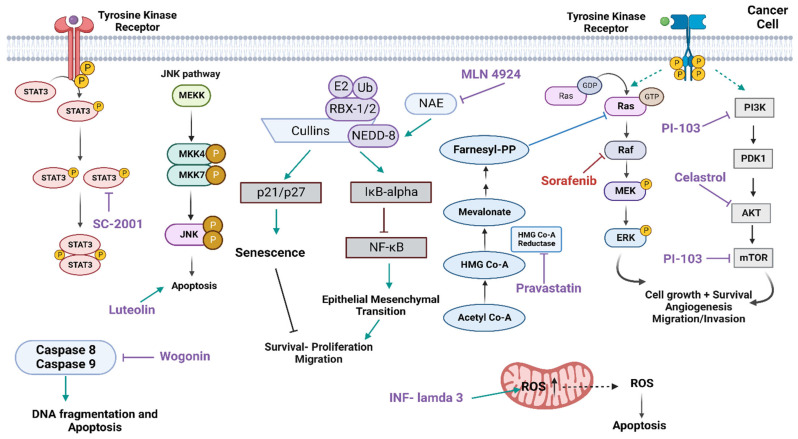
Schematic representation of combinational therapies of sorafenib and other small molecule inhibitors discussed in this review. This simplified schematic does not contain all signal transduction molecules known to be involved in the described signaling cascades but focuses on the targets discussed in this review. This figure was generated by biorender. STAT, Signal transducer and activator of transcription 3; JNK, c-Jun N-terminal kinase; NEDD-8, neural precursor cell expressed developmentally downregulated 8; NF-κB, nuclear factor kappa light chain enhancer of activated B cells; HMG co-A, β-Hydroxy β-methylglutaryl-CoA; RAS, Rat sarcoma virus; MEK, Mitogen-activated protein kinase; ERKs, extracellular signal-regulated kinases; PI3K, Phosphatidylinositol 3-kinase; PDK1, 3-Phosphoinositide-dependent protein kinase 1; AKT, Ak strain transforming; mTOR, mammalian target of rapamycin; INF-lamda 3, Interferon lambda-3; ROS, Reactive oxygen species.

**Figure 5 molecules-27-05537-f005:**
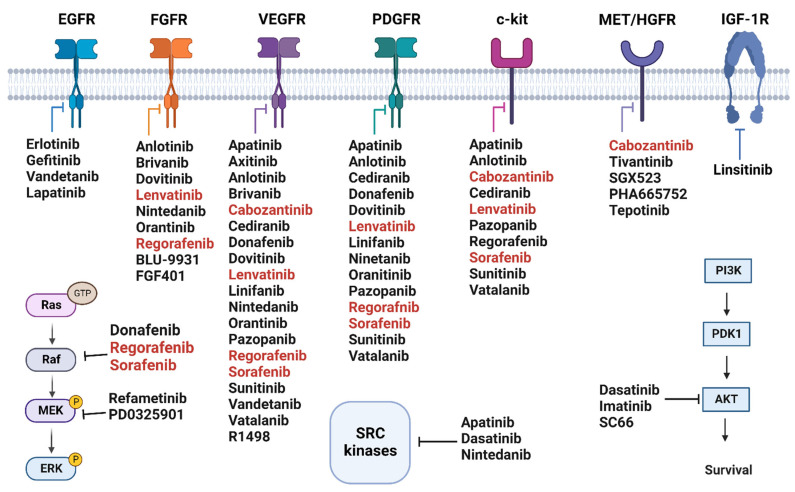
Schematic representations of FDA approved (indicated in red) and not licensed (indicated in black) small molecule inhibitors of tyrosine kinase receptors and other pathways in HCC discussed in this review. This figure was generated by biorender. EGFR, Epidermal Growth Factor Receptor; FGFR, Fibroblast Growth Factor Receptor; VEGFR, Vascular Endothelial Growth Factor Receptor; PDGFR, Platelet Derived Growth Factor Receptor; HGFR, Hepatocyte Growth Factor Receptor; IGF, Insulin Growth Factor Receptor.

**Figure 6 molecules-27-05537-f006:**
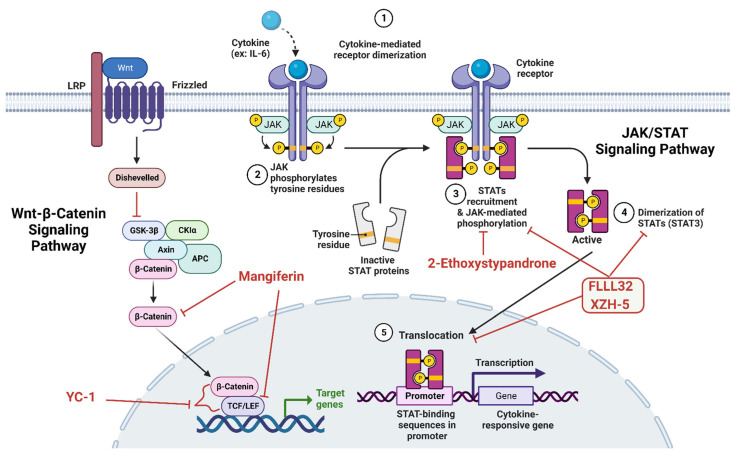
Schematic representations of emerging small molecule inhibitors of Wnt signaling pathway and STAT3 signaling pathway. This figure was generated by biorender. Wnt, Wingless and Int-1; TCF/LEF, T-cell factor/lymphoid enhancer-binding factor; GSK-3β, glycogen synthase kinase-3 beta; APC, Adenomatous Polyposis Coli; LRP, lipoprotein receptor-related protein; JAK, Janus Kinase.

**Table 2 molecules-27-05537-t002:** Small molecule inhibitors and their different targets.

Small Molecule Inhibitor	Target
CMO	P65 protein
APG-1387	Inhibitor of apoptosis proteins (IAPs)
AC-73	CD147
VO-OHpic	Phosphatase and tensin homolog (PTEN)
Rubone	miR34a
FQI1	Transcription factor LSF
AUY922 (luminespib)	Heat shock protein 90 (HSP-90)
Compound 81	Chemokine receptor 6 (CXCR6)
Cambinol	Sirtuin 1 (SIRT-1)
BI 2536	Polo-like kinase 1 (plk-1)
THZ1	cyclin dependent kinase 7 (CDK7)
IPA-3	p21-activated kinase 1 (PAK1)
Alisertib	AURKA

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
