# Peer review of "Small Molecule Inhibitors for Hepatocellular Carcinoma: Advances and Challenges"

_molecules, 2022, doi:10.3390/molecules27175537_

Round 1

Reviewer 1 Report

The manuscript comprehensively describes combinational therapies of Sorafenib and other small molecule inhibitors against HCC. Some improvements need to be made before acceptance. 

1. The methodology followed during the study is described shortly. It would be better to include more detail. The figure is supportive but not easy to understand regarding the selected criteria.

2. Some of the references are given in PubMed links rather than in the journal's suggested reference style.

3. The quality of schematic representations is poor, and needs to be replaced with better versions.

4. The discussion part is very short and general statements are made. The authors need to discuss the advantages and disadvantages of newly described small molecules and the potential of combinatorial strategies for HCC.  

Author Response

1. The methodology followed during the study is described shortly. It would be better to include more detail. The figure is supportive but not easy to understand regarding the selected criteria

Response: Thank you for your comment. More details has been added to the methodology in the manuscript and the supportive figure has been updated. 

2. Some of the references are given in PubMed links rather than in the journal's suggested reference style.

Response: The referencing style has been modified to match that of the journal. 

3. The quality of schematic representations is poor, and needs to be replaced with better versions.

Response: All schematic representations were replaced with higher resolution. 

4. The discussion part is very short and general statements are made. The authors need to discuss the advantages and disadvantages of newly described small molecules and the potential of combinatorial strategies for HCC.  

Response:

We appreciate the reviewer’s comment. More insights were added to the discussion part in the manuscript discussing combinational therapy and current first-line treatment plan for HCC

Reviewer 2 Report

This is an interesting review summerizd on the development of small molecule inhibitors in the treatment of hepatocellular carcinoma (HCC).

The need of this review rises from some general aspect on the summary of target agents for HCC.

The author has also presented results from recent studies about representative small molecules.

This review is well written and comprehensive.

 I have a couple of recommendations.

 The title would be changes as “The development of small molecule inhibitors in the treatment of hepatocellular carcinoma”.

 It would be necessary to change the discussion regarding the recent update in systemic treatment of HCC. Sorafenib is no more first-line treatment modality. See an attached algorithm of HCC treatment.

Author Response

This is an interesting review summarized on the development of small molecule inhibitors in the treatment of hepatocellular carcinoma (HCC).The need of this review rises from some general aspect on the summary of target agents for HCC. The author has also presented results from recent studies about representative small molecules. This review is well written and comprehensive.

Response: We thank the reviewer for his/her kind words about our review and for his/her constructive criticism of it.

1. The title would be changes as “The development of small molecule inhibitors in the treatment of hepatocellular carcinoma”.

Response: We appreciate the reviewer’s comment. We edited the title to be: Small Molecule Inhibitors for Hepatocellular Carcinoma: Advances and Challenges.

2. It would be necessary to change the discussion regarding the recent update in systemic treatment of HCC. Sorafenib is no more first-line treatment modality. See an attached algorithm of HCC treatment.

Response: We thank the reviewer for his valuable comment. The updated first-line treatment of HCC has been added to the discussion section. 
